# Cancer-wide *in silico* analyses using differentially expressed genes demonstrate the functions and clinical relevance of *JAG, DLL,* and *NOTCH*

Jung Yun Kim[1,2], Nayoung Hong[1,2], Seok Won Ham[3], Sehyeon Park[1,2], Sunyoung Seo[1,2], Hyunggee Kim [1,2]*

1 Department of Biotechnology, College of Life Sciences and Biotechnology, Korea University, Seoul, Republic of Korea, 2 Institute of Animal Molecular Biotechnology, Korea University, Seoul, Republic of Korea, 3 MEDIFIC Inc., Hwaseong-si, Gyeonggi-do, Republic of Korea

* hg-kim@korea.ac.kr

**Data Availability Statement:** All TCGA files are available from the cBioPortal database (https://www.cbioportal.org/). All GBM files are available from the GlioVis database (http://gliovis.bioinfo.

## Abstract

Notch ligands [jagged (JAG) and, delta-like (DLL) families] and receptors [NOTCH family] are key regulators of Notch signaling. NOTCH signaling contributes to vascular development, tissue homeostasis, angiogenesis, and cancer progression. To elucidate the universal functions of the JAG, DLL, and NOTCH families and their connections with various biological functions, we examined 15 types of cancer using The Cancer Genome Atlas clinical database. We selected the differentially expressed genes (DEGs), which were positively correlated to the JAG, DLL, and NOTCH families in each cancer. We selected positive and negative hallmark signatures across cancer types. These indicated biological features associated with angiogenesis, hypoxia, KRAS signaling, cell cycle, and MYC targets by gene ontology and gene set enrichment analyses using DEGs. Furthermore, we analyzed single-cell RNA sequencing data to examine the expression of JAG, DLL, and NOTCH families and enrichment of hallmark signatures. Positive signatures identified using DEGs, such as KRAS signaling and hypoxia, were enriched in clusters with high expression of JAG, DLL, and NOTCH families. We subsequently validated the correlation between the JAG, DLL, and NOTCH families and clinical stages, including treatment response, metastasis, and recurrence. In addition, we performed survival analysis to identify hallmark signatures that critically affect patient survival when combining the expression of JAG, DLL, and NOTCH families. By combining the DEG enrichment and hallmark signature enrichment in survival analysis, we suggested unexplored regulatory functions and synergistic effects causing synthetic lethality. Taken together, our observations demonstrate the functions of JAG, DLL, and NOTCH families in cancer malignancy and provide insights into their molecular regulatory mechanisms.

cnio.es/). Datasets of BIC and OV with primary tumors and normal tissues are available from the GEO database (accession numbers GSE65194 and GSE69428). All scRNA-seq data files are available from the GEO database (accession number GSE193884, GSE195665, GSE173278, and GSE125449). All microarray dataset files are available from the GEO database (accession number GSE79671, GSE119937, GSE164359, GSE173855, GSE240829, GSE77940, GSE25694, GSE103479, GSE146163, GSE66698, GSE9600, GSE50509, GSE248830, GSE141174, GSE65021, GSE85258, GSE158408, GSE73168, GSE3325, and GSE151179). Proteomics data for BIC is available from ProteomXchange (accession number PXD008841) Proteomics data for CRC, GBM, HNSC, KRCC, LUAD, OV, and UCEC are available from Proteomics Data Commons (accession number PDC000116, PDC000204, PDC000221, PDC000200, PDC000198, PDC000219, PDC000110, and PDC000125). Proteomics data for LGG is available from https://github.com/derekwong90/LGG_proteomics.

**Funding:** This study was supported by grants from the National Research Foundation of Korea (NRF) funded by the Ministry of Education (2020R1A2C2099668, 2022M3E5F2018255, and RS-2023-00208798 to H. Kim) and the MEDIFIC Inc. (Q2130231). The funders had no role in study design, data collection and analysis, decision to publish, or preparation of the manuscript.

**Competing interests:** H.K. is the founder, majority shareholder, and external director of MEDIFIC Inc. S.W.H. was affiliated with MEDIFIC, Inc. The authors declare no conflict of interest. This does not alter our adherence to PLOS ONE policies on sharing data and materials.

## Introduction

Notch signaling is essential in various developmental processes and in maintaining tissue homeostasis [1–3]. It is involved in modulating multiple downstream pathways, including cell fate decisions during development, cell proliferation, apoptosis, self-renewal, and stem cell differentiation, resulting in distinct physiological and biological outcomes [1–3]. Ligand-receptor interactions initiate canonical Notch signaling, which is highly regulated [1–3].

The four Notch receptor paralogs, NOTCH1, NOTCH2, NOTCH3, and NOTCH4, interact with the Delta-like ligands (DLL1, DLL3, and DLL4) and Jagged (JAG1 and JAG2) protein families [1–3]. Notch receptors have a negative regulatory region that blocks ligand-independent activated Notch signaling [1, 3]. The binding of a ligand leads to a conformational change in Notch receptors, which enables their cleavage mediated by A disintegrin and metalloprotease 10 (ADAM10) or ADAM17/Tumor necrosis factor alpha (TNF-α) converting enzyme (TACE) and γ-secretase [1–3]. After processing the Notch receptors, the Notch intracellular domain (NICD) is released from the plasma membrane to flow freely. The NICD can then enter the nucleus to form a complex with CBF1/Suppressor of Hairless/LAG-1 (CSL) and mastermind-like protein (MAML), which directly regulates the expression of Notch downstream target genes [1–3].

Notch signaling can activate or repress multiple downstream pathways, the intensity of which largely depends on the total amount of NICD [4]. Each type of cancer or clone comprising a tumor can express different combinations of Notch receptors and their ligands, suggesting that dynamic regulation occurs, leading to numerous outcomes [5]. Furthermore, Notch receptors undergo post-translational modifications, especially phosphorylation, which change their affinity for ligands and intracellular half-lives [6–9]. In addition, somatic mutations in Notch genes account for the ligand-independent proteolysis of NOTCH, leading to the upregulation of stable NICD [5, 10].

The dysregulation of Notch signaling plays a crucial role in tumorigenesis and various diseases [4, 10, 11]. The role of NOTCH during tumorigenesis has been described as that of an oncogene; however, opposite functions have also been found as tumor suppressor genes [2, 12]. Tumor propagation by aberrant activation of Notch signaling has been reported in solid cancers of the brain, breast, ovary, lung, skin, and large intestine [5, 12]. However, the tumor-suppressive functions of NOTCH have also been demonstrated in squamous cell carcinoma (SCC) and forebrain glioma [2, 12]. Interestingly, previous studies have suggested that the role of Notch signaling differs depending on the stage of tumor progression; Notch signaling inhibits tumor formation during the early stages, whereas it promotes tumor growth in the late stage [13].

In the past decade, bioinformatics studies have provided evidence suggesting possible connections among biological factors, such as oncogenic alterations, which determine therapeutic response, as demonstrated by bioinformatic analyses [14]. Similarly, another bioinformatics study proposed that somatic copy number alterations in cancer correlate with immune evasion and reduced response to immunotherapy, suggesting that aneuploidy and mutations can be used to predict immune signatures [15]. These studies were usually conducted using resources from public databases that contain information regarding several cancers, such as The Cancer Genome Atlas (TCGA), the International Cancer Genome Consortium (ICGC), and the Catalogue of Somatic Mutations in Cancer (COSMIC) [14–16]. Moreover, developing algorithms for interpreting massive clinical data will allow us to understand the complex network of communication between various factors, which is difficult to elucidate under experimental conditions [14, 17, 18].

In this study, we conducted a cancer-wide *in silico* analysis using the TCGA clinical database to investigate the function and clinical relevance of JAG, DLL, and NOTCH families. We

performed gene ontology (GO) analysis using differentially expressed genes (DEGs) that correlated with the JAG, DLL, and NOTCH families in each type of cancer. We also identified signatures most significantly associated with the JAG, DLL, and NOTCH families. We analyzed single-cell RNA sequencing (scRNA-seq) data, examined the expression of each gene at the cellular level, and identified associated signatures. Finally, we merged the expression and survival data and elucidated possible interactions between genes and signatures responsible for patient prognosis.

## Methods

### Data sources

We used expression datasets from 15 types of cancers, including bladder urothelial carcinoma (BUC), breast invasive carcinoma (BIC), colorectal adenocarcinoma (CRC), glioblastoma (GBM), head and neck squamous carcinoma (HNSC), kidney renal clear cell carcinoma (KRCC), low-grade glioma (LGG), liver hepatocellular carcinoma (LHC), lung adenocarcinoma (LUAD), ovarian serous cystadenocarcinoma (OV), prostate adenocarcinoma (PRAD), skin cutaneous melanoma (SCC), stomach adenocarcinoma (STAD), thyroid carcinoma (TC), and uterine corpus endometrial carcinoma (UCEC).

We downloaded the TCGA datasets containing more than 400 samples from cBioPortal for Cancer Genomics, except for GBM. For GBM, the dataset including both primary tumors and normal tissue samples, which is also from TCGA, was downloaded from GlioVis data portal [19]. Only patients whose gene expression data and overall survival were available in the database were included in our analysis. The comparison of primary tumors and normal tissues was performed based on each TCGA datasets which included the transcriptome of tumor samples and corresponding normal tissues. The datasets of BIC and OV from TCGA provided only tumor samples. To compare tumor and normal in BIC and OV, we obtained additional datasets with the accession numbers GSE65194 and GSE69428 curated by Gene Expression Omnibus (GEO).

Proteomics data for BIC (accession number: PXD008841) was collected from ProteomXchange. We downloaded the proteomics dataset of CRC (accession number: PDC000116), GBM (accession number: PDC000204), HNSC (accession number: PDC000221), KRCC (accession number: PDC000200), LHC (accession number: PDC000198), LUAD (accession number: PDC000219), OV (accession number: PDC000110), and UCEC (accession number: PDC000125) from Proteomic Data Commons. Proteomics data for LGG were collected from https://github.com/derekwong90/LGG_proteomics [20].

BIC, GBM, and LHC scRNA-seq data were collected from the GEO (accession number: GSE195665, GSE193884, GSE173278, and GSE125449).

To compare primary and recurrent tumors, we collected datasets containing expression data of longitudinal biopsies from same patient. Datasets for BIC (accession number: GSE119937), LHC (accession number: GSE164359), HNSC (accession number: GSE173855), OV (accession number: GSE240829), and SCC (accession number: GSE77940) were downloaded from GEO. GBM, LGG, and TC datasets were collected from TCGA.

To confirm the correlation between anti-cancer drug treatment and expression, we collected datasets containing pre- and post-treatment with anti-cancer drugs. Datasets for BIC (accession number: GSE25694), CRC (accession number: GSE103479), KRCC (accession number: GSE146163), LHC (accession number: GSE 66698), HNSC (accession number: GSE9600), and SCC (accession number: GSE50509) were collected from GEO.

We compared the expression of JAG, DLL, and NOTCH families between primary and metastatic tumors using datasets of BIC and LUAD (accession number: GSE248830), CRC

(accession number: GSE141174), HNSC (accession number: GSE65021), KRCC (accession number: GSE85258), LHC (accession number: GSE158408), OV (accession number: GSE73168), PRAD (accession number: GSE3325), and TC (accession number: GSE151179) that collected from GEO. Dataset of SCC were downloaded from TCGA.

The GSE79671 microarray dataset was downloaded from the GEO database. GSE79671 contained the overall survival of patients treated with bevacizumab and standard therapy and the expression data of the tumor samples resected before therapy.

## Selection of the DEGs

In the expression data of each cancer, the genes with an absolute correlation coefficient ($| r |$) value greater than 0.2 for each of the expression levels of the JAG, DLL, and NOTCH families were sorted for the significance analysis of microarrays (SAM). Patients in the dataset were divided into two groups based on the mean ± standard error of the mean (SEM) of JAG, DLL, and NOTCH families' expression levels. SAM analysis was conducted to select genes that were significantly upregulated in each group expressing higher levels of JAG, DLL, and NOTCH families. These sets of genes were referred to as DEGs for the JAG, DLL, and NOTCH families.

## GO analysis

The Database for Annotation, Visualization, and Integrated Discovery (DAVID, version 2023q2) was used to conduct GO analysis of the DEGs compiled using SAM. For our analysis, we collected GO terms belonging to the categories of biological processes, cellular components, molecular functions, and Kyoto Encyclopedia of Genes and Genomes (KEGG) pathways were collected for our analysis. The GO terms with the FDR $q$ values were presented in the results for every tumor type.

## Gene set enrichment analysis (GSEA)

GSEA was performed using cancer hallmark signatures as described previously [21]. The hallmark cancer signatures were downloaded from the Molecular Signatures Database (MsigDB; https://www.gsea-msigdb.org/gsea/msigdb). The enrichment of a signature was significant when the FDR $q$ value was < 0.25 and the normalized enrichment score (NES) was > 1.25.

We performed single-sample GSEA (ssGSEA) using ssGSEA v10.0.1 (https://github.com/GSEA-MSigDB/ssGSEA-gpmodule), using the hallmark signatures and GO terms selected in the DEG-based GO analysis. The sets of genes belonging to the GO terms associated with the JAG, DLL, and NOTCH families were downloaded from AmiGO version 2 (https://soybase.org/amigo/amigo/landing). The enrichment scores from the ssGSEA were normalized to $z$ scores for further analysis. To perform the survival analysis, patients were divided into two groups based on the mean ± SEM of ssGSEA $z$ scores. Heatmaps demonstrating the correlation between the expression or DEG enrichment of JAG, DLL, and NOTCH families and the signature enrichment were generated in Genesis version 2.30.0 (https://github.com/UW-GAC/GENESIS) using the normalized correlation coefficient ($r$).

## Gene network analysis

Gene network analysis was conducted using Cytoscape with the GeneMANIA plugin (version 3.4.1; https://genemania.org/plugin/) as described previously [22]. Eight categories of interactions in the network, including co-expression, physical interaction, genetic interaction, shared protein domains, co-localization, pathway, and predicted interaction, are presented as different colored lines. The relative weights of the connections are described as the thickness of the

lines, and the gray circles denote the essential mediators connecting each component. The interactions present in the network were suggested by merging previous findings and computational predictions. The co-expression category was mostly from the information in GEO, and the categories, including physical interaction, genetic interaction, and pathway, were compiled from previous research and interaction databases, such as BioGRID, Pathway Commons, Reactome, and BioCyc. The shared protein domain category contained information from protein domain databases, including InterPro, SMART, and Pfam. The predicted interaction category demonstrated functional relationships that were verified in other organisms via orthology.

### Analysis of scRNA-seq

Data pre-processing was performed in R, as described in the original scRNA-seq [23]. Normalization was then applied to all cells using the R package "*Seurat*" (v4.3.0) [24]. To reduce the dimensionality of the scRNA-seq data, principal component analysis (PCA) was performed using the *RunPCA* function in *Seurat*. The t-distributed Stochastic Neighbor Embedding (tSNE) was performed using the R package "*tsne*" based on the PCA space using significant PCs. The number of neighbors was set to the same value as *k* used for clustering. Cells were scored according to their similarity to reference cell types using the R package "*SingleR*" (v2.2.0) [25] with the Human Primary Cell Atlas as the main reference. We identified preferentially expressed genes in clusters or DEGs using the *Findallmarker* function in *Seurat*.

### Statistical analyses

All data from the *in vitro* experiments are shown as bar graphs and presented as mean ± SEM. Data were analyzed using a two-tailed Student's t-test ($^*p < 0.05$, $^{**}p < 0.01$, $^{***}p < 0.001$). Error bars in the scatter plots represent the standard error. The Pearson product-moment correlation coefficient (*r*) was calculated using Microsoft Excel. Generation of Kaplan-Meier curves and comparison of overall survival by log-rank t-tests were performed using GraphPad Prism 5 (GraphPad Software, www.graphpad.com).

## Results

### JAG, DLL, and NOTCH families' expression tends to vary across the cancer types

To elucidate the function of JAG, DLL, and NOTCH families, which are associated with various cancers and not limited to any particular cancer type, we accessed the clinical gene expression datasets compiled by TCGA (Fig 1A). We collected expression datasets from 15 types of cancer, including BIC, BUC, CRC, GBM, HNSC, KRCC, LGG, LHC, LUAD, OV, PRAD, SCC, STAD, TC, and UCEC (S1A Fig in S1 File).

We analyzed various aspects of the functions and associated signatures of JAG, DLL, and NOTCH families (Fig 1A and S1B Fig in S1 File). First, we compared the mRNA expression levels of JAG, DLL, and NOTCH families in each cancer type and corresponding normal tissues (Fig 1B and S2A Fig in S1 File). The results showed a diverse expression tendency across cancer types; their expression was either higher or lower than that in normal tissues. For example, the mRNA expression level of *JAG1* was lower in LUAD and UCEC tissue than in normal tissues but was highly expressed in HNSC, KRCC, LGG, SCC, STAD, and TC (Fig 1B). Most of the JAG, DLL, and NOTCH families genes were lower in GBM than in normal brain tissue. However, the proteins were highly expressed in GBM than in normal brain tissue (Fig 1B and S2B Fig in S1 File). In particular, both mRNA and protein expression of NOTCH2 were highly

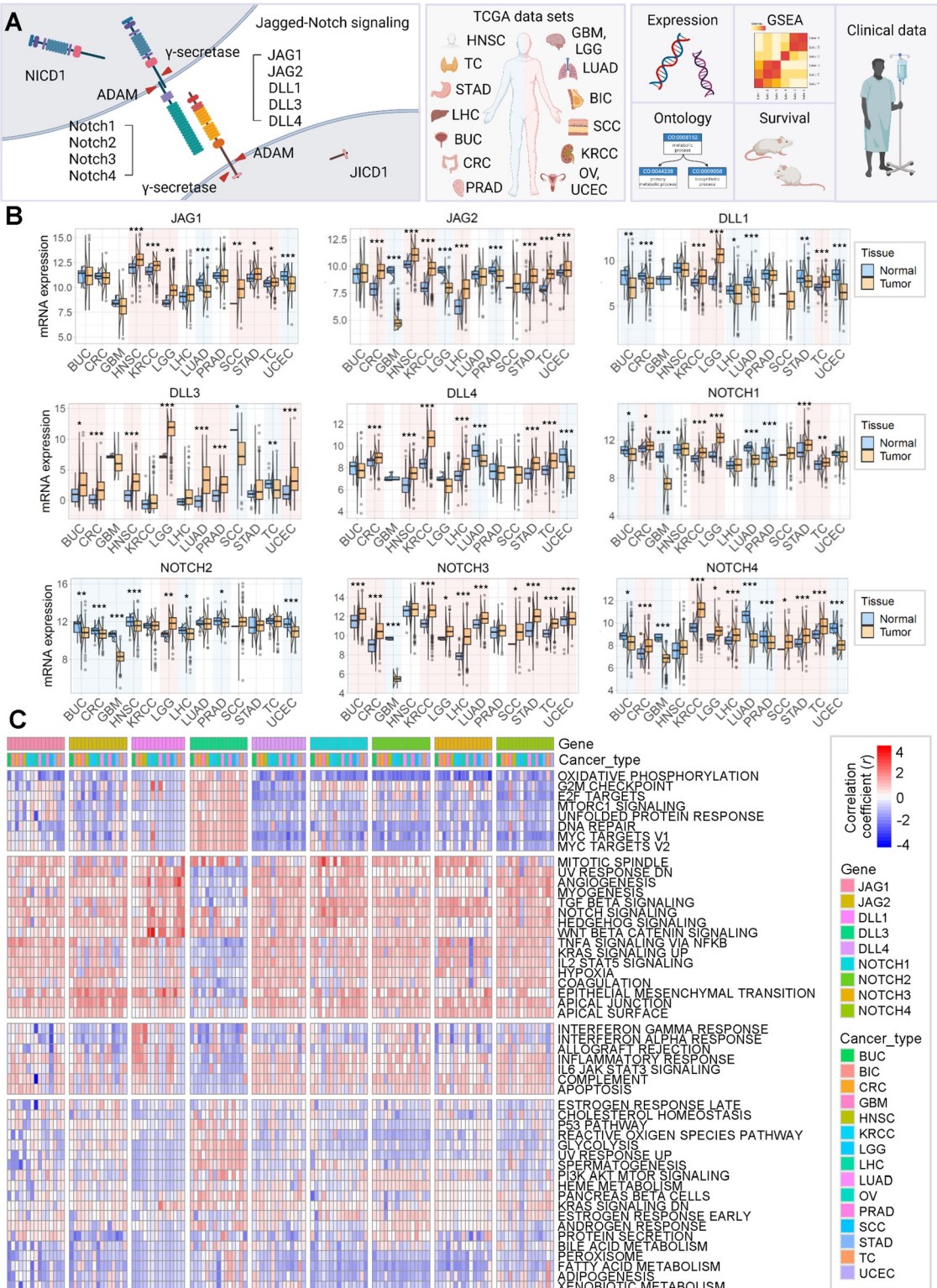

**Fig 1. JAG, DLL, and NOTCH families' expression across the cancer types, and their correlation with hallmark signatures.** (A) Schematic diagram demonstrating the overall process of analysis. This figure was created with [BioRender.com](https://BioRender.com). BUC: bladder urothelial carcinoma, BIC: breast invasive carcinoma, CRC: colorectal adenocarcinoma, GBM: glioblastoma, HNSC: head and neck squamous carcinoma, KRCC: kidney renal clear cell carcinoma, LGG: low-grade glioma, LHC: liver hepatocellular carcinoma, LUAD: lung adenocarcinoma, OV: ovarian serous cystadenocarcinoma, PRAD: prostate adenocarcinoma, SCC: skin cutaneous

melanoma, STAD: stomach adenocarcinoma, TC: thyroid carcinoma, and UCEC: uterine corpus endometrial carcinoma. (B) Box plots comparing the gene expressions of JAG, DLL, and NOTCH families in each cancer type and the corresponding normal tissues. Data were analyzed using a two-tailed Student's t-test (*$p < 0.05$, **$p < 0.01$, ***$p < 0.001$). (C) A heatmap describing the correlation between the hallmark signature enrichment scores and the expression levels of JAG, DLL, and NOTCH families.

expressed only in LGG. The overall expression of JAG, DLL, and NOTCH families genes compared to normal tissues was distinguishable in each cancer. mRNA and protein expression of *JAG1*, *DLL1/4*, and *NOTCH1/4* were lower in LUAD than in normal lung tissue, on the other hand, in KRCC, all the JAG, DLL, and NOTCH families genes except *DLL3* were highly expressed (Fig 1B and S2B Fig in S1 File).

The datasets of BIC and OV from TCGA provided only tumor samples. To give a clear explanation for cancer-specific effects, we conducted further analysis to compare the expression of JAG, DLL, and NOTCH families between tumors and normal tissues. *JAG2*, *DLL3*, *DLL4*, and *NOTCH4* were lower in tumors compared to normal tissues in BIC (S2A Fig in S1 File, left). On the other hand, *JAG2* and *NOTCH3* were highly expressed in tumors, while only *DLL1* was lower in OV than in normal tissues (S2A Fig in S1 File, right).

To validate the correlation between the expression of JAG, DLL, and NOTCH families and tumor progress, we compared the expression according to the tumor stage (S3 Fig in S1 File). Each gene showed a tendency that is different from the tumor stage in each cancer type. For example, *JAG1* increased in a stage-dependent manner in GBM&LGG and SCC and decreased in a stage-dependent manner in UCEC (S3A Fig in S1 File). Most of JAG, DLL, and NOTCH families were stage dependently expressed in GBM (S3 Fig in S1 File). However, there are limited cases showing tumor stage-dependent changes and having higher expression than normal tissues simultaneously.

To validate the molecular signature regulated by the JAG, DLL, and NOTCH families, we ssGSEA using the hallmark signatures from the MSigDB. We investigated the correlation between enrichment scores and the expression of JAG, DLL, and NOTCH families genes (Fig 1C).

All JAG, DLL, and NOTCH families genes, except *DLL3*, showed a similar correlation trend with hallmark signatures across various cancer types (Fig 1C). In particular, there were significantly similar correlations with hallmark signature enrichment in the JAG and NOTCH families. Some signatures, including the G2/M checkpoint and E2F targets, displayed distinguishable correlations depending on the gene and cancer type. However, *DLL3* showed the opposite correlation trend with the hallmark signatures in the JAG and NOTCH families.

Our results demonstrate that JAG, DLL, and NOTCH families gene expression varies among cancer types and that they are differentially associated with molecular signatures depending on the gene and cancer type. The correlations between gene expression and signature enrichment imply that the JAG, DLL, and NOTCH families regulate cellular physiology. However, the variation in expression levels and inconsistencies in correlations depending on the type of cancer impede proper interpretation of the results. Thus, an improved approach is necessary to clearly determine the functions of JAG, DLL, and NOTCH families.

## JAG, DLL, and NOTCH families have common and distinct GO terms across various cancers

ICDs of Notch ligands and receptors were released after cleavages and entered the nucleus, directly regulating the expression of target genes [26–30]. The expression of target genes induces the phenotype resulting from the activation of Notch signaling. We designed a DEG-based cancer-wide *in silico* analysis to determine the functions of JAG, DLL, and NOTCH

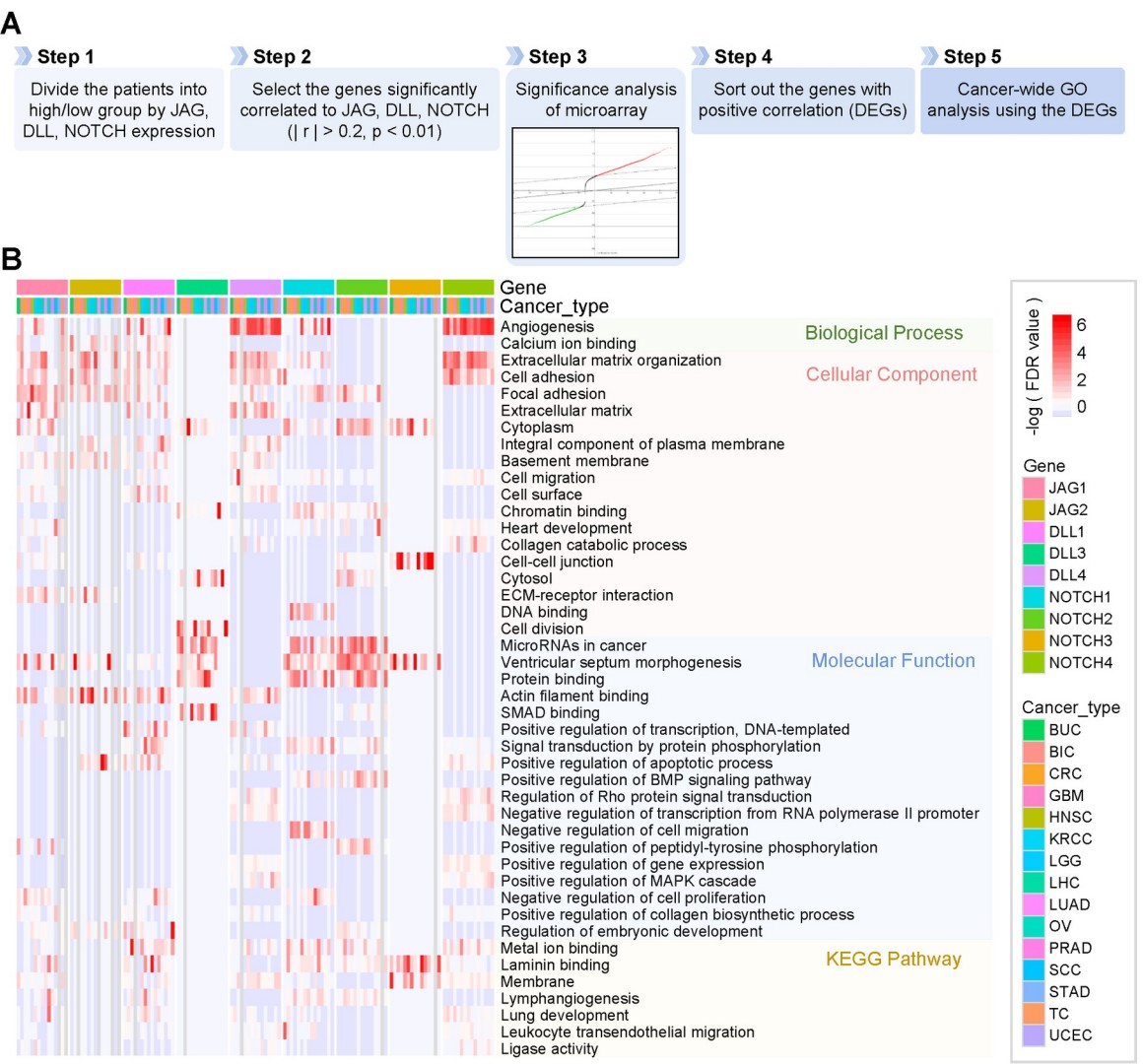

**Fig 2. DEG-based gene ontology (GO) analysis using DEGs from the JAG, DLL, and NOTCH families.** (A) Workflow of GO analysis of DEGs. (B) Heatmap showing the false discovery rate (FDR) *q*-values of core GO analysis results. Each category of the GO terms includes biological processes, cellular components, molecular functions, and the KEGG pathway.

families using clinical databases (Fig 2A). We divided the patients with each cancer type into two groups based on the mRNA expression levels of the JAG, DLL, and NOTCH families. The genes for which the absolute value of the correlation coefficient (| *r* |) for the JAG, DLL, and NOTCH families was greater than 0.2 were processed for SAM analysis. Using the results from the analysis, we compiled the genes showing significant positive correlations with the DEG for each JAG, DLL, and NOTCH families; for instance, a set of genes from the SAM analysis with *JAG1* in the BUC dataset is termed JAG1-DEG in BUC.

Next, we compared the DEG enrichment for each JAG, DLL, and NOTCH family according to the tumor stage (S4 Fig in S1 File). The expression of JAG, DLL, and NOTCH families was correlated with tumor stage in a few cancer types, while each DEG enrichment was significantly associated with most cancers (S3, S4 Figs in S1 File).

Interestingly, the DEG enrichment of JAG1, DLL4, and all Notch receptors increased according to tumor stage (S4 Fig in S1 File). NOTCH1-DEG enrichment increased

significantly despite the decrease in mRNA expression of NOTCH1 according to tumor stage (S3, S4 Figs in S1 File). The transcriptional DEGs of JAG1 and NOTCH1 overlap significantly according to our previous results [29]. Our results might be due to the ligand-dependent signaling activation of NOTCH1 by abundant JAG1 and DLL4. Furthermore, because DEG of Notch ligands and receptors highly overlap, upregulated JAG1 and DLL4 might have driven these trends.

For KRCC, DEG enrichment of most JAG, DLL, and NOTCH families decreased stage-dependently while only DLL3-DEG enrichment increased (S4 Fig in S1 File). These results suggested that DLL3 may have an inverse effect on KRCC as opposed to others. In addition, the DEG of all Notch ligands increased in stage-dependent manner in LHC.

The diversity of the composition and expression of DEGs for each gene and tumor influenced the different trends in DEG enrichment for each gene. Moreover, the Notch signal interacts with other oncogenic signals, and they also affect the regulation of the expression of target genes of the Notch signal [31, 32]. Since the signaling mechanisms activated in each tumor are different, the aspect of their DEG enrichment is observed in various trends [33, 34].

We then conducted a GO analysis with the DEGs from each cancer type and listed the GO terms with the lowest sum of false discovery rate (FDR) from the analysis for each cancer type (Fig 2B and S5, S6 Figs in S1 File). The outcomes from the cancer-wide DEG-based GO analysis demonstrated biological features associated with the gene, which are universal across cancers; therefore, they might provide clues to deduce the gene's function. We performed a GO analysis of the remaining genes belonging to the JAG, DLL, and NOTCH families. Notably, all JAG, DLL, and NOTCH families genes, except *DLL3* and *NOTCH3*, showed significant correlations with extracellular matrix (ECM) organization, cell migration, and adhesion, strongly implying that the JAG, DLL, and NOTCH families could play a crucial role in regulating cell motility and cancer invasion (Fig 2B and S5, S6 Figs in S1 File).

The GO analysis conducted with the JAG1-DEGs showed that *JAG1* was associated with various functions, including cell adhesion, focal adhesion, actin filament binding, cell-matrix adhesion, integrin signaling pathway, membrane components, and signatures that critically regulate cell motility and cancer invasiveness (Fig 2B and S5, S6 Figs in S1 File). JAG1-DEGs were mainly composed of genes that encoded proteins whose subcellular localizations were the site of contact and the scaffold of the cell, such as focal adhesions, cell-cell junctions, cytoskeletons, and membrane rafts (Fig 2B and S5, S6 Figs in S1 File). *JAG1* was also correlated with hypoxia, angiogenesis, and Transforming Growth Factor β (TGFβ) signaling (Fig 2B and S5, S6 Figs in S1 File).

JAG2-DEGs contained genes related to cell adhesion and ECM but showed fewer GO terms compared to the JAG1-DEGs, which might imply that the impact of JAG2 on modulating physiological functions and signatures is less than that of JAG1 (Fig 2B and S5, S6 Figs in S1 File).

Each DLL family demonstrated distinguishable results in DEG-based GO analysis (Fig 2B and S5, S6 Figs in S1 File). Both *DLL1* and *DLL4* were associated with cell motility, ECM organization, hypoxia, and angiogenesis, and their DEGs were mostly localized in the plasma membrane and regions of cell-to-cell or cell-to-ECM interactions; however, they also displayed different GO terms (Fig 2B and S5, S6 Figs in S1 File). In particular, DLL1-DEGs included genes responsible for transcriptional regulation and DNA binding. In contrast, *DLL4* showed a much stronger correlation with the regulation of blood vessel formation and endothelial development, as inferred from the number of GO terms (Fig 2B and S5, S6 Figs in S1 File). In addition, the molecular function of the DLL1-DEGs was transcriptional regulation by DNA binding, whereas DLL4-DEGs were likely to act as transducers of cellular signaling (Fig 2B and S5, S6 Figs in S1 File). DLL3-DEGs were associated with mRNA splicing and cell division;

however, there were fewer GO terms, possibly because of the small size of the DEGs in the SAM analysis (Fig 2B and S5, S6 Figs in S1 File).

The GO analysis of NOTCH1-DEGs demonstrated a possible correlation between *NOTCH1* and transcriptional modulation (Fig 2B and S5, S6 Figs in S1 File). Associations between NOTCH1 and GO terms, such as positive and negative regulation of transcription, chromatin binding, and sequence-specific DNA binding, corresponded to NOTCH-mediated signaling pathway mechanisms, which were already known (Fig 2B and S5, S6 Figs in S1 File). This is consistent with previous findings showing that the NOTCH1 intracellular domain remodels chromatin and recruits transcription factors to regulate downstream genes [1–3]. Furthermore, NOTCH1-DEGs were associated with angiogenesis, cell migration, and ECM organization, as well as JAG1-DEGs (Fig 2B and S5 and S6 Figs in S1 File). Because JAG1 is known to bind to NOTCH1, these features may be modulated by NOTCH1 upon a signal driven by JAG1 binding. However, it is also possible that JAG1 and NOTCH1 regulate their features independently. Correspondingly, NOTCH1-DEGs contained genes encoding nuclear proteins different from JAG1-DEGs, which denotes a distinct function mode regulating these features (Fig 2B and S5, S6 Figs in S1 File).

Biological processes related to NOTCH2-DEGs included transcriptional regulation, cell migration, and vascular development, as well as NOTCH1-DEGs (Fig 2B and S5, S6 Figs in S1 File). However, proteins encoded by NOTCH2-DEGs were localized in all subcellular compartments, and the functions of the proteins were more diverse than those encoded by NOTCH1-DEGs, which bind to transcriptional factors, chromatin, and metal ions and regulate cell-to-cell adhesion and kinase activity, affecting various cellular signaling pathways (Fig 2B and S5, S6 Figs in S1 File). GO analysis using NOTCH3-DEGs showed that relatively fewer GO terms were affected by *NOTCH3* (Fig 2B and S5, S6 Figs in S1 File). *NOTCH3* correlated with the cytoplasm, cell-cell junctions, ventricular septum morphogenesis, laminin-binding, and membrane (Fig 2B and S5, S6 Figs in S1 File). NOTCH4-DEGs were associated with biological processes such as angiogenesis, cell motility, ECM organization, and hypoxia, of which the encoded proteins mainly existed in the membrane and ECM, harboring functions such as calcium ion binding, integrin binding, and regulation of cellular signaling (Fig 2B and S5, S6 Figs in S1 File). Notably, we found considerable similarity between the results of the GO analysis using JAG1-DEGs and NOTCH4-DEGs, suggesting a possible correlation that still needs to be elucidated.

## DEGs of JAG, DLL, and NOTCH families are essential modulators of the hallmark signatures

We investigated the correlation between the expression levels of JAG, DLL, and NOTCH families and the enrichment scores of the hallmark signatures using Gene Set Enrichment Analysis (GSEA) (Fig 1C). Our DEG-based GO analysis provided a list of GO terms associated with each JAG, DLL, and NOTCH family member (Fig 2B and S5, S6 Figs in S1 File). Using these results, we identified the most significant biological signatures regulated by the JAG, DLL, and NOTCH families to understand their implications in cellular physiology and clinical outcomes.

To identify the cellular correlations of the JAG, DLL, and NOTCH families, we calculated the correlation coefficient between the expression levels or the DEG enrichment scores of the JAG, DLL, and NOTCH families and the enrichment scores of the signatures sorted from the GSEA and GO analysis (Fig 3A). We selected those signatures with a positive sum of *r* through clustering, including apical surface, epithelial-mesenchymal transition (EMT), coagulation, hypoxia, Interleukin-2-signal transducer and activator of transcription 5 (IL2-STAT5)

**A**

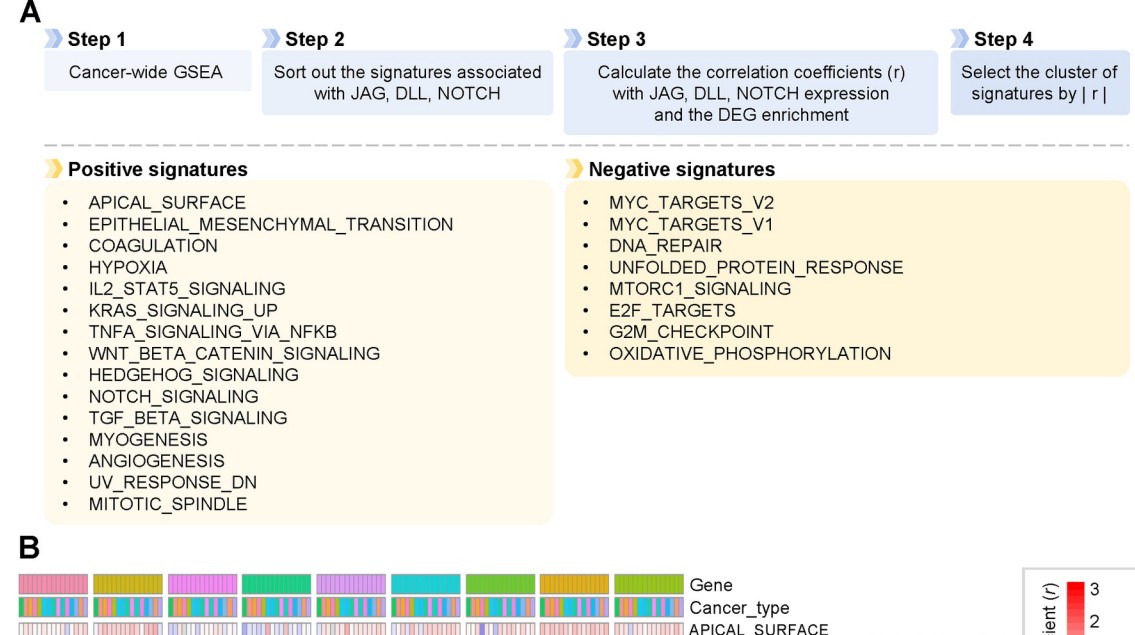

**B**

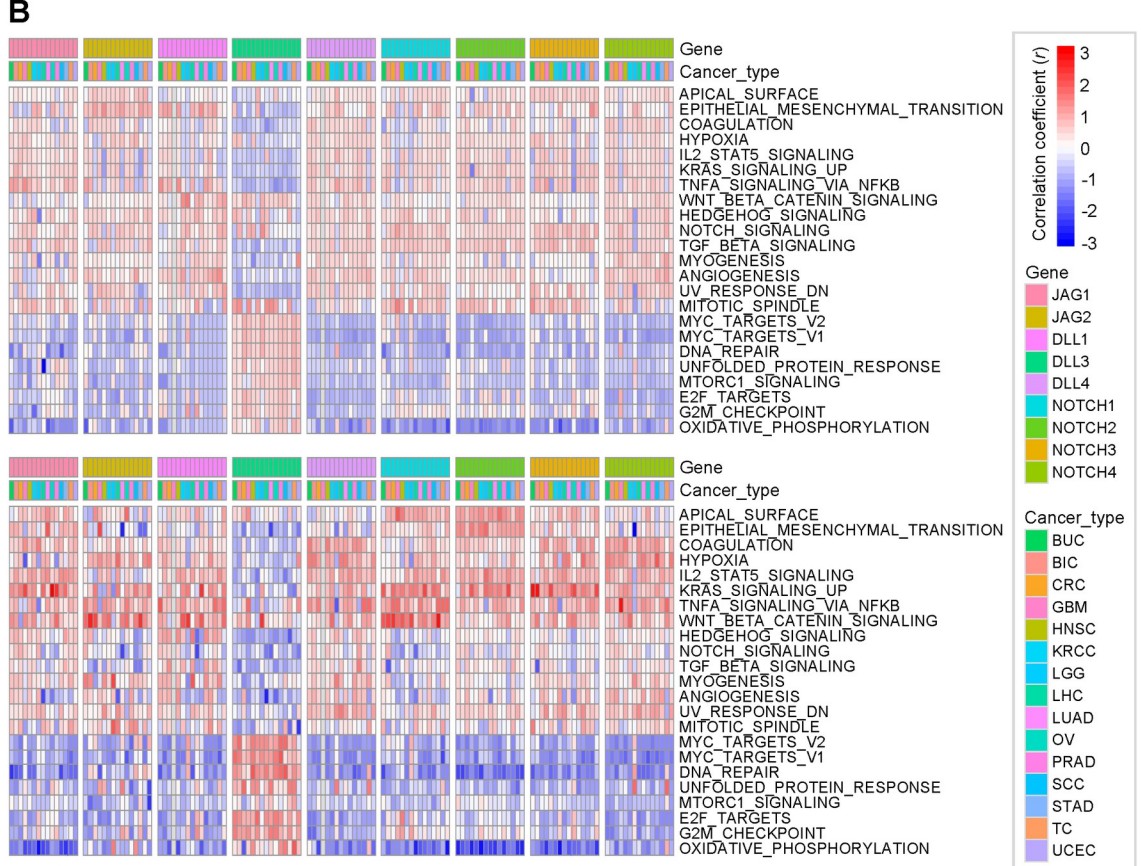

**Fig 3. Selection of the signatures that are the most significantly associated with JAG, DLL, and NOTCH.** (A) A schematic diagram describing the process for selecting positive and negative signatures is listed below. (B) Heatmap showing the correlation between the hallmark signatures and expression levels (top) or DEG enrichment scores (bottom) of the JAG, DLL, and NOTCH families.

signaling, KRAS signaling UP, TNF-α signaling via Nuclear factor κB, Wnt β catenin signaling, hedgehog signaling, NOTCH signaling, TGFβ signaling, myogenesis, angiogenesis, UV response DN, and mitotic spindle (hereafter, positive signatures, Fig 3A). Similarly, we selected signatures with a negative sum of *r* as follows: MYC target V2, MYC target V1, DNA repair, unfolded protein response, mammalian target of rapamycin complex 1 (mTORC1) signaling, E2F targets, G2/M checkpoint, and oxidative phosphorylation (hereafter, negative signatures, Fig 3A). Positive signatures generally correlated positively with the expression levels and DEG enrichment scores of the JAG, DLL, and NOTCH families. In contrast, negative signatures displayed a negative overall correlation, except for *DLL3*, which showed an opposite trend (Fig 3B). We speculated that the notably high significance of the positive and negative signatures implied that the JAG, DLL, and NOTCH families were essential modulators of the signatures and that the signatures could be upstream regulators of the JAG, DLL, and NOTCH families.

We then investigated how the DEGs interacted with each other and how the JAG, DLL, and NOTCH families were involved in the network to modulate various biological functions. We conducted gene network analysis using the Cytoscape GeneMANIA plugin with the DEGs selected in GBM to validate the interactions between the JAG, DLL, and NOTCH families and other genes in regulating each signature (S7 Fig in S1 File). The analysis provided a map of the interactions between the DEGs by integrating information from previous studies. The types of interactions and relative weights of the networks are represented by the colors and thicknesses of the lines, respectively. The mediators not included in the DEGs but essential for the interactions are marked with gray circles, the size of which denotes the interaction score with the DEGs. Gene network plots corresponding to signatures, including NOTCH signaling, angiogenesis, hypoxia, and KRAS signaling, are presented in S7 Fig in S1 File.

## JAG, DLL, and NOTCH families are associated with hypoxia and KRAS signaling in single-cell and cancer-wide

To verify the correlations between the JAG, DLL, and NOTCH families and the signatures demonstrated by *in silico* analysis, we analyzed the scRNA-seq data of GBM [23]. Visualization of GBM cells using tSNE revealed extremely high heterogeneity, and the cells were divided into 13 clusters (Fig 4A).

The expression of JAG, DLL, and NOTCH families was characterized in each cluster (Fig 4B and S8A Fig in S1 File). The JAG family exhibited high expression in cluster 10, whereas the DLL family showed no specific cluster with high expression. *JAG1* was highly expressed in clusters 2, 3, 5, and 10, with the highest expression observed in cluster 5. Except for *NOTCH4*, all NOTCH family members were commonly expressed in clusters 0, 1, and 6. JAG, DLL, and NOTCH family members were highly expressed in distinct clusters.

To identify the biological processes in clusters with high JAG, DLL, and NOTCH families' expression, we selected highly expressed DEGs in each cluster. GO analysis revealed that the DEGs of cluster 5, which highly expressed *JAG1*, were associated with biological processes, such as cell adhesion, extracellular matrix organization, and translation (Fig 4C). Interestingly, GO analysis conducted on the DEGs of cluster 10, which highly expressed the JAG family, showed significant similarity to cluster 5 (Fig 4C and S8B Fig in S1 File). The DEGs of cluster 0, which generally expressed the NOTCH family, were associated with translation, ribosomes, and cytoplasm. Some GO terms were similar to clusters 5 and 10, but GO terms related to cytoplasm and ribosomes were distinct from other clusters (Fig 4C and S8B, S8C Fig in S1 File). However, GO analysis of cluster DEGs revealed limited GO terms.

GSEA demonstrated that clusters 5 and 10 were enriched with the signatures belonging to the positive signatures, such as KRAS signaling UP, UV response UP, and hypoxia (Fig 4D).

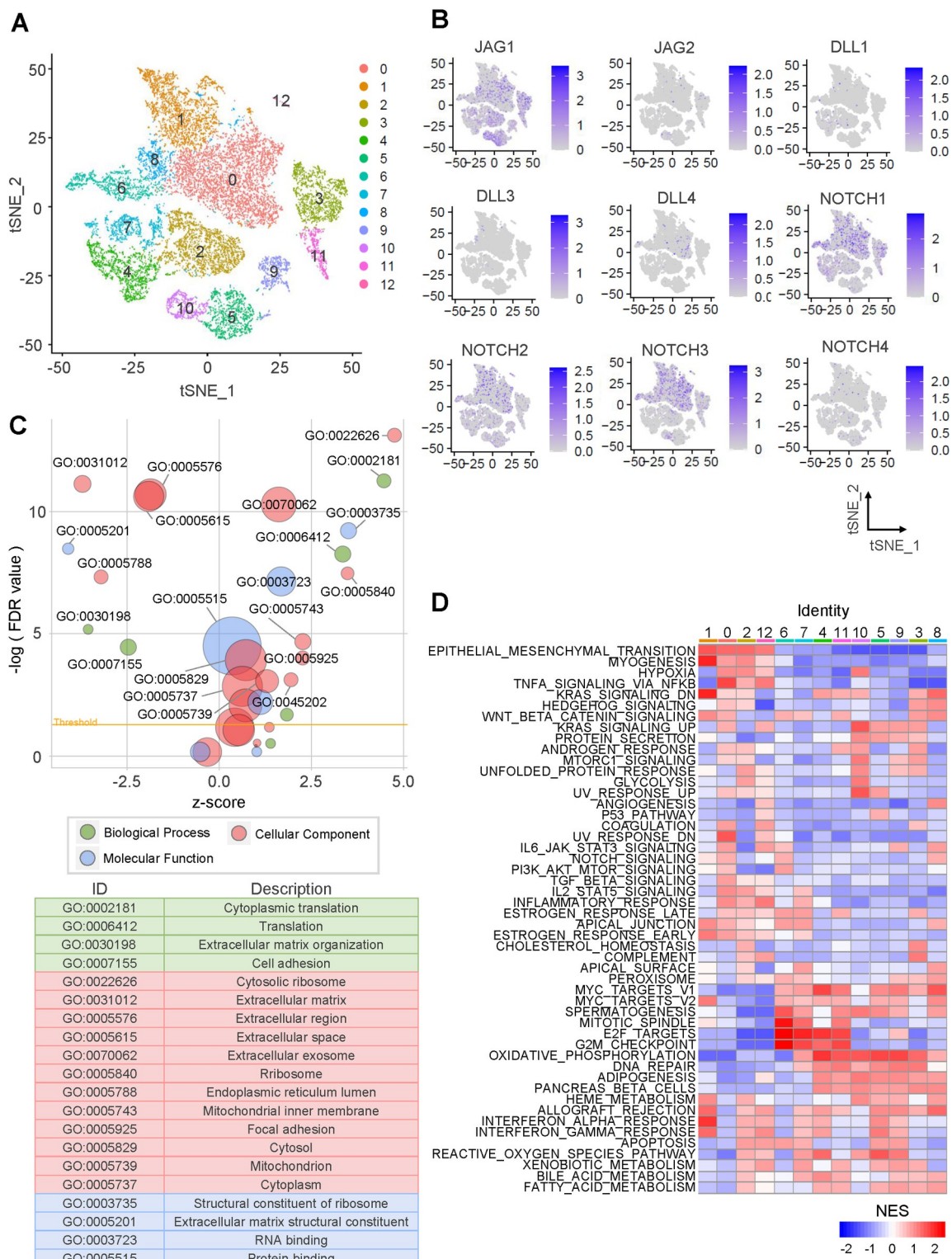

**Fig 4. Expression of the JAG, DLL, and NOTCH families and enrichment of positive and negative signatures using scRNA-seq of GBM.** (A) tSNE plot of GBM cells colored by cluster. (B) Log-normalized expression of JAG, DLL, and NOTCH families. (C) Bubble plot demonstrating GO term enrichment in Cluster 5. The bubble size represents the number of DEGs in each GO term. GO terms [-log (FDR value) < 2.5] are listed in the table below. (D) Heatmap showing the enrichment of hallmark signatures in each cluster.

However, most positive signatures were enriched in cluster 0, including EMT, myogenesis, hypoxia, KRAS signaling UP, NOTCH signaling, and IL2-STAT5 signaling (Fig 4D). While the NOTCH family regulates major biological processes, the JAG and DLL families regulate only a tiny fraction of the biological process but are independent.

Additionally, we analyzed scRNA-seq data of BIC, GBM, and LHC to verify consistency at various cancer types (S9, S10 Figs in S1 File) [23, 35, 36]. There were no clusters in which the JAG, DLL, and NOTCH families were individually expressed. Among Notch ligands, *JAG1* was most expressed in many clusters, and the expression of receptors showed different patterns across cancer types. Specifically, multiple Notch ligands and receptors are expressed together in most clusters of LHC (S9B and S9C Fig in S1 File). In other words, members of the JAG, DLL, and NOTCH families were expressed in distinct clusters.

Most clusters with high JAG, DLL, and NOTCH families were enriched with positive signatures such as hypoxia, KRAS signaling, UP, EMT, NOTCH signaling, myogenesis, angiogenesis, and TGFβ signaling. These GSEA results were consistent with Figs 3B and 4D.

Hypoxia is a prominent feature of the tumor microenvironment induced by oxygen deprivation, often caused by oncogene-mediated rapid enlargement of the tumor bulk without vascularization. Furthermore, vigorous Mitogen-activated protein kinase (MAPK) signaling driven by gain-of-function mutations in *KRAS* is a common genetic feature of cancer. In our *in silico* analysis, the JAG, DLL, and NOTCH families exhibited evident correlations with hypoxia and KRAS signaling (Fig 3B). Likewise, scRNA-seq analysis demonstrated that these signatures were modulated by the JAG, DLL, and NOTCH families (Fig 4 and S8 Fig in S1 File).

## JAG, DLL, and NOTCH families each correlate with different clinical stages of distinct tumor types

While the gene expression of JAG, DLL, and NOTCH families themselves did not show a significant correlation with tumor stage (as shown in S3 Fig in S1 File), a considerable number of hallmark signatures were enriched in patients with highly expressed JAG, DLL, and NOTCH families (Figs 1C and 3B). As the hallmarks of cancer promote tumor progression, we performed further analyses to link the transcriptomes of the JAG, DLL, and NOTCH families to tumor progression.

We compared the expression of JAG, DLL, and NOTCH families in primary and metastatic tumors (Fig 5A and S11 Fig in S1 File). Expression changes in JAG and DLL families varied among genes and cancer types. For example, metastatic KRCC tumors decreased the expression of *JAG1* and *DLL4*. Metastatic PRAD tumors increased the expression of *JAG2*, *DLL3*, and *DLL4* but *JAG1* decreased. However, the expression of the NOTCH family between primary and metastatic tumors differed only in a few cases (S11B-S11E Fig in S1 File).

Furthermore, we compared the expression between samples from patients before and after anti-cancer therapy (Fig 5B, 5C and S12 Figs in S1 File). The expression of JAG, DLL, and NOTCH families changed before and after anti-cancer therapy in some cancers. Although restricted to BIC and HNSC, the expression of most JAG and DLL families increased after anti-cancer therapy. Most genes whose expression changed upon therapy showed gradual changes in HNSC and KRCC as the treatment progressed (Fig 5C and S12C Fig in S1 File).

Additionally, we analyzed the expression of JAG, DLL, and NOTCH families in primary and recurrent tumors from longitudinal biopsies of the same patient (Fig 5D, 5E, S13, S14 Figs in S1 File). Expression of the JAG and DLL families increased in most patients with recurrent BIC, LHC, OV, SC, and TC. However, the NOTCH family expression in these patients showed diverse patterns without any tendency. There were no differences in the expression of JAG, DLL, and NOTCH families between primary and recurrent tumors in GBM, LGG, and HNSC.

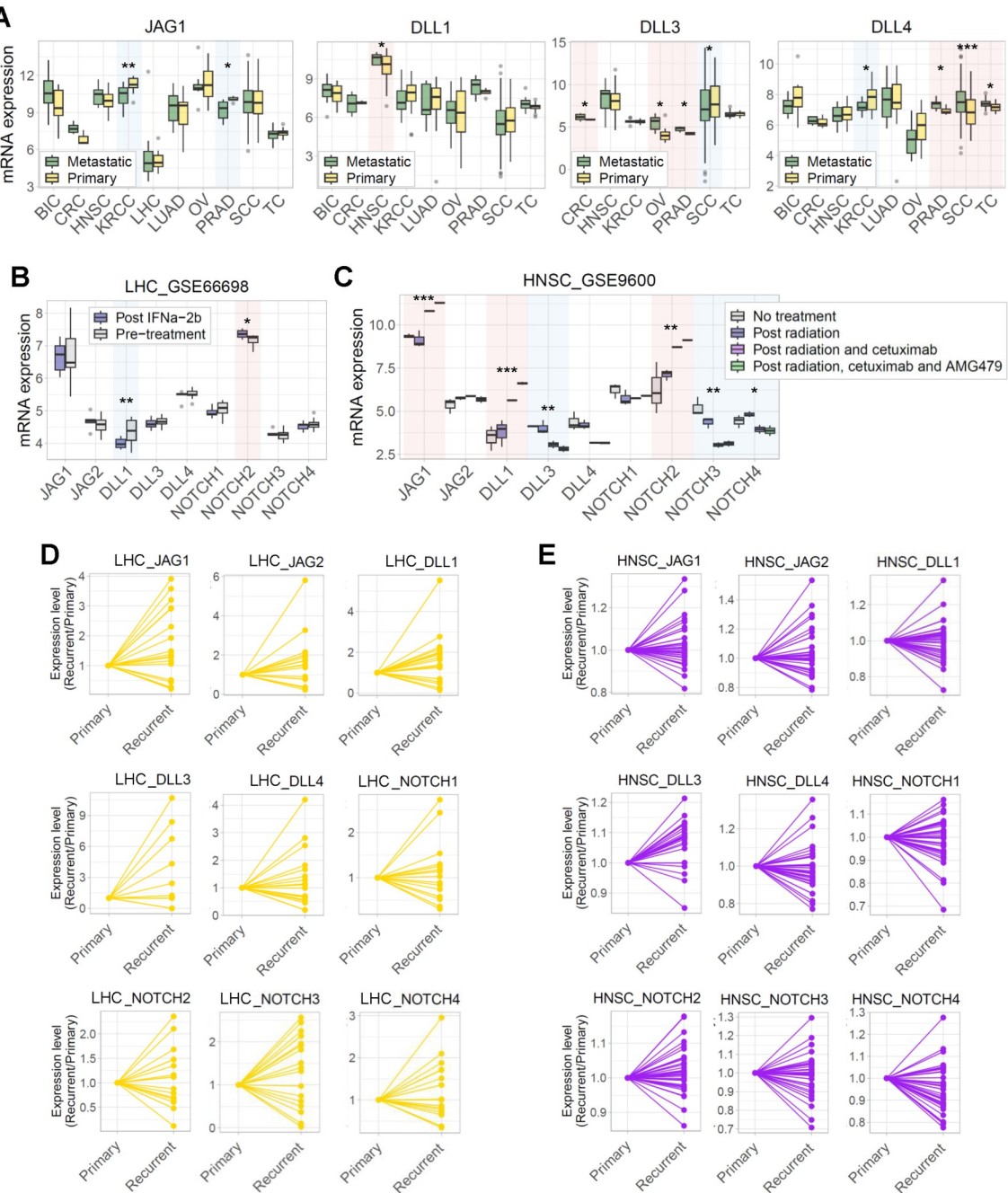

**Fig 5. Correlation between the expression of JAG, DLL, and NOTCH families and clinical and pathological characteristics.** (A) Box plots comparing the expression of *JAG1*, *DLL1*, *DLL3*, and *DLL4* in primary and metastatic tumors. Data were analyzed using a two-tailed Student's t-test (*$p < 0.05$, **$p < 0.01$, ***$p < 0.001$). (B) Box plots comparing the expression of JAG, DLL, and NOTCH families before and after chemotherapy with IFNa-2b in LHC. Data were analyzed using a two-tailed Student's t-test (*$p < 0.05$, **$p < 0.01$). (C) Box plots comparing the expression of JAG, DLL, and NOTCH families before and after radiotherapy and chemotherapy with cetuximab and AMG479 in HNSC. (D, E) Scatter plots showing the expression of JAG, DLL, and NOTCH families between primary and recurrent cancers of each patient with (D) LHC and (E) HNSC.

Our findings align with prior research on the correlation between the expression of JAG, DLL, and NOTCH families and clinical stages. For instance, DLL3 is more highly expressed in metastatic CRC tumors than in primary lesions, and its diagnostic role with prognostic value

and therapeutic potential for anti-cancer therapy has been suggested [37]. Similarly, DLL4 has been linked to invasion and metastasis, with its expression in TC holding prognostic significance [38]. Moreover, the expression of JAG1 and DLL4 correlates with time to relapse in LHC [39].

To confirm the relationship between DEG for each gene and clinicopathological features in common solid tumors, we performed additional analyses to compare DEG enrichment with four different clinicopathological features, including tumor stage, metastasis, recurrent, and post-chemotherapy status of tumors.

Expression of JAG, DLL, and NOTCH families between primary and metastatic tumors differed in most cancers, and DEG enrichment was more closely related to tumor stage than the expression of ligands and receptors (Fig 5A, S3, S4, S11, S15, and S16 Figs in S1 File). However, DEG enrichment between primary and metastatic tumors showed differences only in a few. Even in the case of PRAD, the expression of JAG1 and JAG1-DEG enrichment were reversed. The expression of JAG1 decreased, but DEG enrichment increased in metastatic tumors (Fig 5A and S15A in S1 File).

For each cancer type, we compared the DEG enrichment of each gene between samples from patients before and after anti-cancer therapy (S17 Fig in S1 File). Gene expression and DEG enrichment are similar only in BIC and HNSC (S17A and S17C in S1 File). There were no differences in DEG enrichment for each gene before and after anti-cancer therapy in CRC, KRCC, and LHC (S17B, S17D, and S17E Fig in S1 File).

Both expression and DEG enrichment of JAG1, DLL1, and NOTCH4 significantly increased after anti-cancer therapy in BIC (S17A Fig in S1 File). In addition, JAG2, NOTCH1, and NOTCH3 tend to increase after anti-cancer therapy, although there is no significance. Interestingly, DLL3 decreased after anti-cancer therapy, contrary to DEG enrichment of other genes.

In HNSC, both expression and DEG enrichment of JAG1, DLL1, and NOTCH2 significantly increased after anti-cancer therapy, while DLL3 and NOTCH4 significantly decreased (S17C Fig in S1 File). However, NOTCH3 expression and NOTCH3-DEG enrichment showed opposite trends in HNSC. There were no differences in all gene expression before and after anti-cancer therapy in SCC, but JAG1-, JAG2-, and NOTCH2-DEG enrichment significantly increased or decreased after anti-cancer therapy (S17F Fig in S1 File). DEG enrichment can be significantly changed depending on activating or inhibiting other signals that interact with the Notch signaling after anti-cancer therapy [31, 32].

We compared the DEG enrichment in primary and recurrent tumors (S18, S19 Figs in S1 File). Changes in DEG enrichment between primary and recurrent tumors were limited to BIC, LGG, and GBM. In Notch ligands, only DLL3-DEG enrichment decreased in recurrent BIC and JAG1-, JAG2-, and DLL1-DEG enrichment decreased in recurrent GBM (S18 Fig in S1 File). DEG enrichment of Notch receptors was also similar (S19 Fig in S1 File). NOTCH1- and NOTCH2-DEG enrichment decreased in recurrent BIC. NOTCH4-DEG enrichment decreased in recurrent GBM, and NOTCH2 DEG decreased in recurrent LGG.

Notch signaling is a cell-cell interaction-dependent mechanism known to regulate gene expression in a cell-type and context-dependent manner. Notch signaling promotes and inhibits tumor development in a context-dependent manner [33, 34]. In addition, the outcomes of the Notch signal can be altered as Notch receptors crosstalk with other signal receptors in the tumor microenvironment. Because our DEGs for each gene were selected based on the primary tumor, the enrichment of their DEGs cannot confirm a context-dependent effect in metastatic or recurrent tumors.

Collectively, these findings suggest that Notch signaling may serve as a predominant regulatory mechanism throughout tumor progression, metastasis, resistance to therapy, and

recurrence corroborating previous studies [40, 41]. Additionally, the receptor-independent activation of the JAG and DLL families may influence tumor aggressiveness [28, 30, 42]. In addition, our results hint at a previously unexplored relationship between Notch signaling, therapeutic effects, and the potential for recurrence.

## Combination of the expression or DEG enrichment with the signature enrichment improves patient survival analysis

Using *in silico* analysis, we compiled DEGs and selected signatures strongly associated with the JAG, DLL, and NOTCH families. Our results demonstrated relationships between the genes and signatures, and these relationships imply that the gene critically modulates or synergizes with the signature, or *vice versa*, affecting cancer progression and recurrence. Therefore, we hypothesized that combining the expression of JAG, DLL, and NOTCH families or their DEGs with signature enrichment scores makes predicting patient prognosis based on their gene expression profiles. We compared the overall survival of patients divided into groups based on the expression of JAG, DLL, and NOTCH families and signature enrichment scores (Fig 6 and S20-S23 Figs in S1 File).

Patients were divided into high- and low-expression groups in the clinical datasets according to the JAG, DLL, and NOTCH families' expression. The DEG enrichment scores, signature (positive and negative) enrichment scores, and overall survival were compared (Fig 6 and S20-S23 Figs in S1 File). Survival analysis using expression and DEG enrichment demonstrated distinguishable results; for example, in KRCC, only *JAG1* expression led to better outcomes in patients with KRCC; however, the analysis with DEG enrichment showed that all the DEGs led to longer survival, except for the DLL3-DEGs (Fig 6A). Not only did DLL3 show the opposite DEG enrichment tendency to others, but its hallmarks enrichment tendency and effect on patient survival also showed an opposite trend (Figs 3B, 6A, S4 Fig in S1 File). In addition, *JAG1* expression did not affect clinical outcomes, whereas the enrichment of DEGs negatively impacted on survival in GBM (Fig 6A). Similarly, survival analysis using the enrichment scores of the positive and negative signatures revealed the effect of the signatures on patient survival (S20 Fig in S1 File).

We then analyzed whether a combination of expression or DEG enrichment with signature enrichment in the survival analysis would explain the relevance of the relationship between a gene and a signature, for instance, unexplored regulatory functions and synergy causing synthetic lethality. Furthermore, by comparing the results of our analysis with those of previous studies, it is possible to understand context-dependent outcomes from the cooperation of signals. Thus, we divided the patients into four groups depending on the expression levels (or the DEG enrichment scores) and signature enrichment scores and conducted a survival analysis (Fig 6B–6D and S21-S23 Figs in S1 File).

Survival analysis using the expression of JAG and NOTCH families and angiogenesis signature enrichment in GBM demonstrated that in the group expressing higher levels of these genes, angiogenesis signature enrichment negatively impacted patient survival (Fig 6B and S21A Fig in S1 File). In addition, the survival analysis results using the DEGs and signature enrichment elucidated the relationships between JAG and NOTCH families, hypoxia, and KRAS signaling (Fig 6C and S21-S23 Figs in S1 File). DEG enrichment in the JAG and NOTCH families was also associated with a similar patient prognosis to that analyzed with gene expression (Fig 6C and S22A Fig in S1 File). We compared these results to the outcomes from a clinical study providing information regarding the survival of patients with GBM who received bevacizumab combined with standard therapy and the gene expression profiles of the tumor before treatment (Fig 6D). Survival analysis revealed that patients with higher *JAG1* or

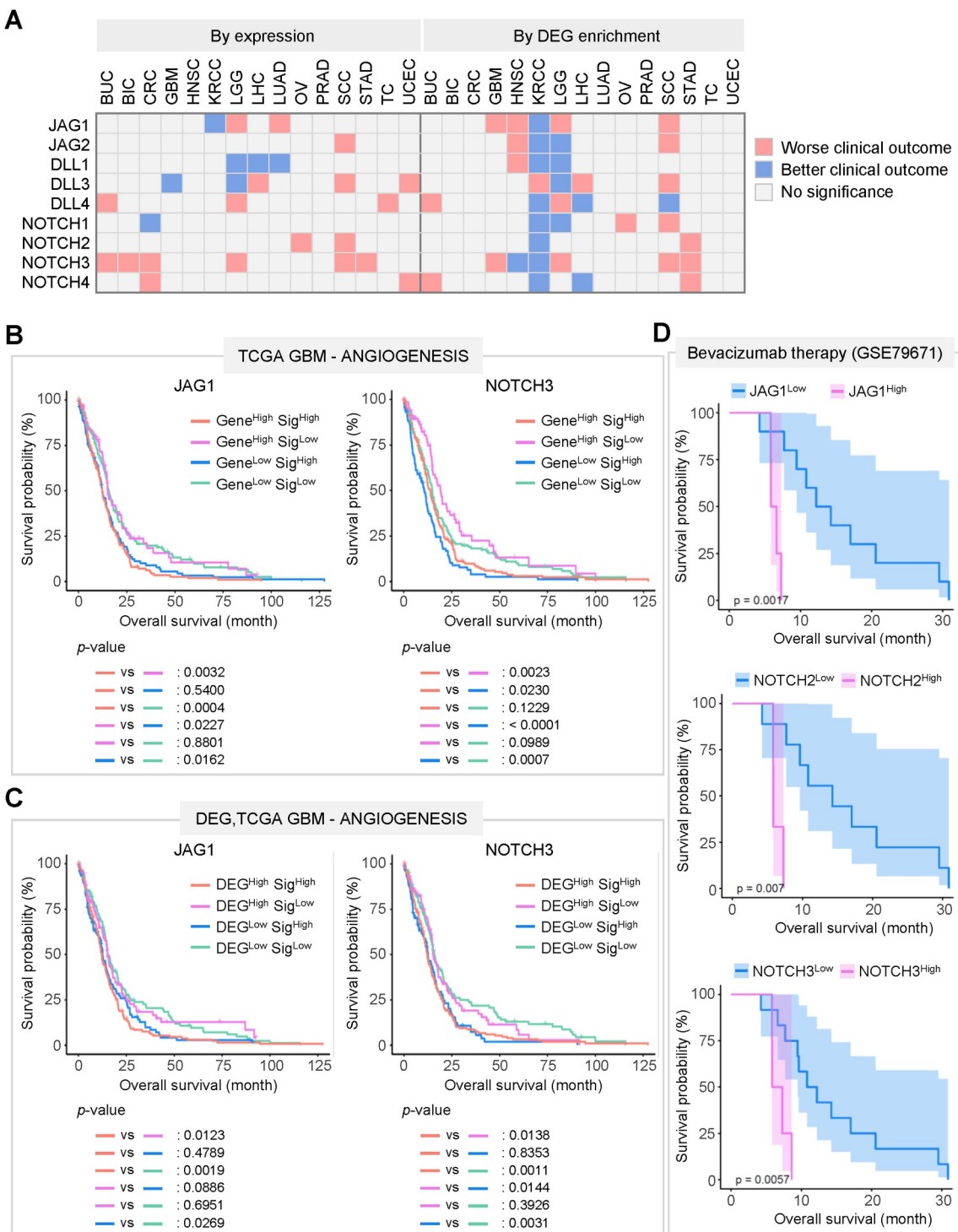

**Fig 6. Survival analysis suggests the clinical relevance of JAG, DLL, and NOTCH families and the associated signatures.** (A) A matrix demonstrating the overall result from the survival analysis using the expression levels or DEG enrichment scores of JAG, DLL, and NOTCH families in the datasets of each cancer type. (B) Kaplan-Meier survival plots showing the overall survival of patients with GBM, considering the expression of *JAG1* or *NOTCH3* and enrichment of angiogenesis signature. (C) Kaplan-Meier survival plots demonstrating the patients' prognostic outcomes concerning the DEG enrichment of JAG1 and NOTCH3 and enrichment of angiogenesis signature. (D) Kaplan-Meier survival plots comparing the overall survival of the patients with GBM treated with bevacizumab in combination with conventional anti-cancer therapy by the expression of *JAG1*, *NOTCH2*, and *NOTCH3*.

*NOTCH2* or *NOTCH3* expression did not respond to therapy (Fig 6D). Considering that the patients in the TCGA GBM dataset were not treated with bevacizumab, it is likely that the cooperation of JAG1, NOTCH2, NOTCH3, and angiogenesis affected prognosis differently, depending on the situation.

In KRCC, a group of the patients showing relatively higher enrichment of TGFβ signaling signature exhibited more prolonged survival when JAG1- or NOTCH1-DEGs were enriched (S23 Fig in S1 File). Notably, a previous study on the mouse kidney tubular cells suggested that inductions of Jag1 and Hey1 are necessary for EMT triggered by TGFβ-Smad signaling [43]. Although cancer invasion and metastasis promoted by EMT are the typical features of malignant tumors, our analysis revealed that in the patients with enriched TGFβ signaling, lower enrichment of JAG1- or NOTCH1-DEGs led to worse prognosis. The result implies that concurrent activation of JAG1 or NOTCH1 downstream pathway with TGFβ signaling is important in the process of invasion and metastasis; however, it might be disadvantageous for the progression of the bulk of the tumor.

Our analysis also provides evidence of the possible relationships not proposed in previous studies. We observed that in patients with KRCC with a lower DNA repair signature, the enrichment of JAG1-DEG resulted in better clinical outcomes (S23 Fig in S1 File). In addition, among patients with PRAD with an enriched oxidative phosphorylation signature, the group with relatively lower NOTCH2-DEG enrichment scores exhibited more prolonged survival (S23 Fig in S1 File). These results demonstrate the possible interaction between DEGs and signatures affecting cancer progression, which should be validated by further research.

## Discussion

The implications of JAG, DLL, and NOTCH signaling in developmental processes and cancer progression have been suggested using numerous experimental approaches. Moreover, targeting NOTCH signaling as an anti-cancer therapy has undergone several clinical trials with promising outcomes. However, little is known about the associations of JAG, DLL, and NOTCH families with signaling pathways and their physiological functions in tumor tissues, which are responsible for cancer characteristics and patients' prognosis. Furthermore, previous studies on the roles of the JAG, DLL, and NOTCH families have primarily been performed in a single type of tumor or tissue; thus, it is necessary to elucidate their common function across different cancer types.

Through *in silico* analysis and scRNA-seq, we found that the JAG, DLL, and NOTCH families were significantly associated with cancer signatures, including hypoxia, angiogenesis, and KRAS signaling up. Many studies on various cancer types have discussed the mechanisms of the modulatory interactions among these signatures. Hypoxia-inducible factor 1α (HIF1α) and HIF2α, stabilized by hypoxic stress, are the most prominent transcriptional activators of pro-angiogenic factors such as nitric oxide synthases and vascular endothelial growth factor (VEGF) [44]. Oncogenic RAS mutations also facilitate angiogenesis via the MAPK; in particular, the MEK-ERK pathway directly induces the expression of VEGF and fibroblast growth factor 2 (FGF2), which are also up-regulated indirectly by the JNK-JUN-COX2 pathway [45]. In addition, mutant KRAS enhances the hypoxia responses by increasing HIF1α translation through the MAPK-mTOR signal [45]. Likewise, receptors for advanced glycation end products (RAGE) are expressed in response to hypoxic stimuli and interact with mutant KRAS to sustain the activation of the KRAS signaling pathway [46].

Following our results, several studies have suggested a connection between the NOTCH signaling pathway and signatures, including hypoxia response, angiogenesis, and KRAS signaling [47]. NOTCH is a regulator of sprouting angiogenesis that simultaneously promotes tumor

vascularization in the presence of ligands expressed by neighboring tumor cells [47–49]. It was also demonstrated that MAPK signaling facilitates angiogenesis in HNSC by upregulating *JAG1* expression to trigger NOTCH signaling in the surrounding endothelial cells [49].

According to the cancer-wide GO and GSEA results, the JAG, DLL, and NOTCH families had different enriched GO terms and hallmark signatures (Figs 2–4). In addition, by analyzing the scRNA-seq data of GBM, the expression of each family was confirmed in each cluster, which was enriched with distinct GO terms and hallmark signatures. These results suggest that the JAG and DLL families, which are limited to Notch ligands, have independent functions. Correspondingly, several studies have revealed the NOTCH-independent function of the JAG1 intracellular domain (JICD1) in regulating biological events and suggested an inhibitory function against NOTCH signaling [50–53]. Mesenchymal stem cells in the bone marrow overexpressing JICD1, but not JAG1, showed increased expression of stromal cell-derived factor 1 (SDF1) and promoted the proliferation of CD34$^+$ hematopoietic progenitor cells in co-culture condition [50]. Genetic mutations responsible for the impaired cleavage of JAG1 have been found in patients with Alagille syndrome [51]. Our previous study showed that JICD1 binds to chromatin modifiers, which promotes the expression of genes related to GBM propagation [29].

In the present study, we demonstrated the cancer-wide function of the JAG, DLL, and NOTCH families and discussed the relevance of the associated signatures in determining cancer progression and therapeutic resistance in patient tumors. Using SAM analysis, we selected DEGs positively correlated with the JAG, DLL, and NOTCH families. However, the negatively correlated genes were not sorted through the analysis, possibly because of difficulties in quantifying the extremely low mRNA expression. Moreover, since our *in silico* analysis focused on correlation rather than direct interaction, the detailed regulatory mechanism should be defined by experimental approaches. Taken together, a comprehensive investigation of the role of the JAG, DLL, and NOTCH families, considering the relevance of their association with various biological features, would contribute to a deeper understanding of the molecular regulatory mechanisms of cancer progression, thereby identifying the most efficient therapeutic modalities for these patients.

## Supporting information

**S1 File. S1-S23 Figs.**
(PDF)

## Acknowledgments

We thank all members of the Cancer Growth Regulation Lab for their support and technical assistance. This study was supported by the School of Life Sciences and Biotechnology of Korea University.

## Author Contributions

**Conceptualization:** Jung Yun Kim, Hyunggee Kim.

**Formal analysis:** Jung Yun Kim.

**Investigation:** Nayoung Hong, Seok Won Ham, Sehyeon Park, Sunyoung Seo.

**Supervision:** Hyunggee Kim.

**Writing – original draft:** Jung Yun Kim.

**Writing – review & editing:** Nayoung Hong, Seok Won Ham, Sehyeon Park, Sunyoung Seo, Hyunggee Kim.

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
