## [Decision Letter · Decision Letter 0]

9 Jan 2024

PONE-D-23-37633Cancer-wide In Silico Analyses Using Differentially Expressed Genes Demonstrate the Functions and Clinical Relevance of JAG, DLL, and NOTCHPLOS ONE

Dear Dr. Kim,

Thank you for submitting your manuscript to PLOS ONE. After careful consideration, we feel that it has merit but does not fully meet PLOS ONE’s publication criteria as it currently stands. Therefore, we invite you to submit a revised version of the manuscript that addresses the points raised during the review process.

We look forward to receiving your revised manuscript.

Kind regards,

Kenji Tanigaki, Ph.D., M.D.

Academic Editor

PLOS ONE

 [This study was supported by grants from the National Research Foundation of Korea (NRF) funded by the Ministry of Education (2020R1A2C2099668, 2022M3E5F2018255, and RS-2023-00208798 to H. Kim) a grant funded by MEDIFIC Inc. (Q2130231).].  

[We thank all members of the Cancer Growth Regulation Lab for their support and technical assistance. This study was supported by grants from the National Research Foundation of Korea (NRF) funded by the Ministry of Education (2020R1A2C2099668, 2022M3E5F2018255, and RS-2023-00208798 to H. Kim) a grant funded by MEDIFIC Inc. (Q2130231), and a KU internal grant from the School of Life Sciences and Biotechnology of Korea University.]

 [This study was supported by grants from the National Research Foundation of Korea (NRF) funded by the Ministry of Education (2020R1A2C2099668, 2022M3E5F2018255, and RS-2023-00208798 to H. Kim) a grant funded by MEDIFIC Inc. (Q2130231).]. 

[H.K. is the founder, majority shareholder, and external director of MEDIFIC Inc. S.W.H. was affiliated with MEDIFIC, Inc. The authors declare no conflicts of interest.]. 

5. We note that Figure 1A in your submission contain copyrighted images. All PLOS content is published under the Creative Commons Attribution License (CC BY 4.0), which means that the manuscript, images, and Supporting Information files will be freely available online, and any third party is permitted to access, download, copy, distribute, and use these materials in any way, even commercially, with proper attribution. For more information, see our copyright guidelines: http://journals.plos.org/plosone/s/licenses-and-copyright.

a. You may seek permission from the original copyright holder of Figure 1A to publish the content specifically under the CC BY 4.0 license. 

Reviewers' comments:

Reviewer's Responses to Questions

**Comments to the Author**

1. Is the manuscript technically sound, and do the data support the conclusions?

Reviewer #1: Partly

Reviewer #2: Partly

2. Has the statistical analysis been performed appropriately and rigorously? 

Reviewer #1: Yes

Reviewer #2: Yes

3. Have the authors made all data underlying the findings in their manuscript fully available?

Reviewer #1: No

Reviewer #2: Yes

4. Is the manuscript presented in an intelligible fashion and written in standard English?

Reviewer #1: Yes

Reviewer #2: Yes

5. Review Comments to the Author

Reviewer #1: 1.Using in silico analysis, the investigator demonstrat the functions and clinical relevance of JAG, DLL, and NOTCH. But the data are only genetic data and survival data. There is a lack of information on other clinical and pathological characteristics of patients, which can not reflect the relationship between gene expression and disease in a panoramic way.

2.The study included as many as 15 cases of tumor species, and the expression of these genes was different in different tumor species and different clinical stages, and their functions were inconsistent, which could not give readers a clear message, express a clear idea, and lack of reference value for follow-up studies.

3.It is suggested that on the basis of this study, we should focus on one or several tumor species for more in-depth analysis and obtain more clear conclusions.

4. Please provide clear pictures and diagrams as required by the journal.

Reviewer #2: Title: Cancer-wide in silico analyses using differentially expressed genes demonstrate the functions and clinical relevance of JAG, DLL, and NOTCH

Major Comments:

A. Methods:

Data download and processing:

1. Justification for dataset selection: It would be helpful if the authors provided a brief rationale for selecting these specific datasets. This information would enhance the readers understanding of the dataset suitability and reinforce the relevance of the subsequent analyses. Furthermore, reason for using GlioVis data portal instead of TCGA GBM datasets for the study.

2. Justification for not using Normal tissue datasets: It will give a clear picture if author analyses the normal tissue dataset which can help to distinguish better between cancer-specific effects.

3. Justification for not evaluating the effects on additional clinical outcomes like treatment response, metastasis, recurrence: Analysis of just single factor like overall survival is a limiting factor of this study and does decrease the generalizability of the study.

4. Explanation on using only mRNA expression data: it would be helpful if author will incorporate proteomic and metabolic data for clear insights.

B. Results:

1. Explanation: In second paragraph of result section, author mentioned comparing normal tissue expression with cancer tissue, but they have not mentioned using normal tissue sample for study and datasets they have downloaded. Author should justify the reason and should mention the source of dataset collection.

2. In the 5th paragraph under the subheading “JAG, DLL, and NOTCH family’s expression tend to vary across the cancer types”, using proteomics datasets/protein expression would help the author to coordinate the JAG, DLL, and NOTCH families regulation in cancer.

3. Clarification: why did not author performed cross-validation using independent datasets to prevent overfitting and conduct subgroup analysis to account for patient heterogeneity which could have decreased the inconsistency in correlations observed.

4. Justification: Author analysis single cell sequencing data which provides only a snapshot. But Heterogeneity within tumour samples was not addressed, performing spatial and temporal modelling with multi-region sequencing and circulating tumour DNA data will strengthen the analysis.

Minor comments:

1. There are many typing and grammatical errors, throughout the whole paper including figure and figure legends.

2. Few abbreviations with the full forms are not mentioned.

3. Figure legends should provide more details.

4. The lack of description for why certain analyses were chosen leaves the rationale unclear.

6. PLOS authors have the option to publish the peer review history of their article (what does this mean?). If published, this will include your full peer review and any attached files.

Reviewer #1: No

Reviewer #2: No

---

## [Author Response · Author response to Decision Letter 0]

14 Feb 2024

Reviewer #1: 

1. Using in silico analysis, the investigator demonstrates the functions and clinical relevance of JAG, DLL, and NOTCH. But the data are only genetic data and survival data. There is a lack of information on other clinical and pathological characteristics of patients, which can not reflect the relationship between gene expression and disease in a panoramic way.

Response: Thank you for your insightful comment. To reflect the relationship between the gene expression of the JAG, DLL, and NOTCH families and clinical/pathological characteristics, we performed additional analyses. The analyses include comparisons between primary/metastatic, pre-/post-chemotherapy, and primary/recurrent tumor.

First, we investigated the gene expression of the JAG, DLL, and NOTCH families in primary or metastatic tumors (Response Figure 1). Changes in expression of the JAG and DLL families were diverse across each gene and cancer type. For example, JAG2, DLL3, and DLL4 were increased in the metastatic tumors of PRAD, whereas JAG1 decreased. In contrast, transcriptomic changes in the NOTCH family have only been observed in some cases.

Second, we compared expression between samples from patients before and after therapy (Response Figure 2). Although restricted to BIC and HNSC, the expression of NOTCH family did not change or decrease, whereas that of JAG and DLL increased (Response Figure 2B and 2C).

We also performed an analysis between primary and recurrent tumors in eight cases of tumor species (Response Figure 3-5). The samples were restricted to longitudinal biopsies from the same patient. In GBM, LGG, and HNSC, there were no differences in the expression of JAG, DLL, and NOTCH families between primary and recurrent tumors. In contrast, several patients with BIC, LHC, OV, SCC, and TC showed increased expression of JAG and DLL families without decline. However, the expression of NOTCH family in these patients showed diverse patterns without any tendency. Additionally, the expression of most of NOTCH family was slightly different, whereas that of JAG and DLL increased in several tumor types, such as CRC, HNSC, OV, PRAD, SCC, and TC.

Collectively, these results indicate that ligand-dependent NOTCH signaling regulation might play a predominant role among the various regulatory mechanisms involved in tumor progression, drug resistance, recurrence, and metastasis, consistent with findings from previous studies (1-3). In addition, receptor-independent activation of the JAG and DLL family may affect aggressiveness of tumor as previously reported (4-6).

2.The study included as many as 15 cases of tumor species, and the expression of these genes was different in different tumor species and different clinical stages, and their functions were inconsistent, which could not give readers a clear message, express a clear idea, and lack of reference value for follow-up studies.

Response: Thank you for your constructive comment. Previously, comprehensive analyses of JAG, DLL, and NOTCH expression and their features associated with prognostic values, such as correlation with patient survival and biological pathways, were performed for individual tumor types (7-10). However, each result was analyzed under different analytic conditions. Thus, we focused on analyzing these genes under identical conditions and suggesting common or distinctive features of the JAG, DLL, and NOTCH families based on 50 cancer-hallmark gene sets. 

Additionally, we investigated whether the expression levels of the JAG, DLL, and NOTCH families increased with tumor progression reflected by tumor stages (Response Figure 6). Stage-dependent upregulation of expression was observed for several genes including JAG1 in GBM, LGG, and SCC. In contrast, in UCEC, the expression of JAG1 decreased with tumor progression. Similarly, each gene exhibited a different trend in transcriptomic changes for each cancer type. However, there are limited cases showing tumor stage-dependent changes and simultaneously higher expression than in normal tissue.

Analyses of gene expression patterns according to clinical stages, such as pre-/post-therapy, primary/recurrent, and primary/metastatic tumors, demonstrated that the expression of JAG, DLL, and NOTCH families have diverse trends in each tumor and clinicopathological classification.

For example, DLL1 was upregulated in metastatic and post-therapy tumors compared to that in primary and pre-therapy tumors, whereas there was no difference after recurrence in HNSC (Response Figure 7A and 7B). These results indicate that DLL1 may be correlated with tumor metastasis and drug resistance, suggesting that DLL1 is a potential therapeutic target. In addition, although the expression of DLL1 was not correlated with patient survival, the DLL1-DEGs enrichment was significantly correlated with a worse prognosis (Response Figure 7C). Furthermore, patient outcome was correlated with the enrichment of hallmark signatures, including hypoxia, KRAS signaling, unfolded protein response, and mTORC1 signaling (Response Figure 7D). 

Moreover, the expression of JAG and NOTCH families was associated with patient prognosis, showing an enriched angiogenesis signature in GBM. The finding corresponds to responsiveness to combinatorial bevacizumab therapy with standard therapy (Response Figure 8).

Overall, while establishing a direct correlation between the stage-dependent expression of JAG, DLL, and NOTCH families and tumor progression remains challenging, our findings provide insight into predicting therapeutic response and the likelihood of recurrence in targeted therapy against Notch signaling.

3.It is suggested that on the basis of this study, we should focus on one or several tumor species for more in-depth analysis and obtain more clear conclusions.

Response: Thank you for your suggestion. We conducted an analysis to assess the correlations between the hallmark signatures selected with positive or negative sum or r and the expression levels or DEG enrichment scores of the JAG, DLL, and NOTCH families. The results showed that the expression of JAG, DLL, and NOTCH family genes, except DLL3, was positively correlated with positive signatures and vice versa in most cancer types (Response Figure 9). Also, we investigated the global network involving the JAG, DLL, and NOTCH families and their DEGs which modulate various biological functions (Response Figure 10). Gene network analysis revealed that JAG, DLL, and NOTCH families were associated with genes regulating various signatures including Notch signaling, angiogenesis, hypoxia, and KRAS signaling. Collectively, these findings suggest an intricate interplay among the JAG, DLL, and NOTCH families, enabling them to collectively orchestrate common features across diverse tumor types. 

Furthermore, patient survival analyses were conducted based on the gene expression levels of the JAG, DLL, and NOTCH families, DEG enrichment, and signature enrichment (Response Figure 11). Survival analysis based on gene expression, DEG enrichment scores, and signature enrichment individually showed different trends in accordance with each gene or cancer type. 

We also compared patient outcomes according to the combination of gene expression or DEG enrichment with signature enrichment. The correlation with patient survival varied according to the combination and tumor type. For example, in GBM, patients with higher expression of JAG and NOTCH families or higher enrichment of DEGs with enriched angiogenesis signature showed a worse prognosis (Response Figure 12 and 13). These results were reflected in the responsiveness to combination therapy of bevacizumab with standard therapy in patients with GBM, according to the expression levels of JAG1, NOTCH2, and NOTCH3. 

In KRCC, patients with higher enrichment of JAG1- or NOTCH1-DEGs combined with an enriched TGFβ signaling signature showed a better prognosis (Response Figure 14). Additionally, higher JAG1-DEGs enrichment with a lower DNA repair signature led to better clinical outcomes (Response Figure 14). Additionally, according to the combinations of gene expression or DEG enrichment with signature enrichment such as DLL4 with E2F target in BIC and gene expression or DEGs of NOTCH2 and NOTCH3 with apical junction in CRC, patient survival varied among different cancer types, suggesting an independent function for each gene in certain cancer type (Response Figure 14). 

Taken together, our results suggest that JAG, DLL, and NOTCH families genes interact with each other to modulate various biological features in a complex network and regulate specific features separately. Furthermore, through the analysis of correlations between the transcriptome, hallmark signature, and clinical relevance in patient cohorts, our results provide valuable insights for identifying specific phenotypes as therapeutic targets and predicting the therapeutic effects of specific drugs in individual patient. 

4. Please provide clear pictures and diagrams as required by the journal.

Response: We apologize for the inconvenience caused by the figures that did not meet the journal’s criteria. We have revised all figures and diagrams, as well as adjusted the font size in accordance to the guidelines.

Reviewer #2: Title: Cancer-wide in silico analyses using differentially expressed genes demonstrate the functions and clinical relevance of JAG, DLL, and NOTCH

Major Comments:

A. Methods:

Data download and processing:

1. Justification for dataset selection: It would be helpful if the authors provided a brief rationale for selecting these specific datasets. This information would enhance the readers understanding of the dataset suitability and reinforce the relevance of the subsequent analyses. Furthermore, reason for using GlioVis data portal instead of TCGA GBM datasets for the study.

Response: Thank you for the suggestion. The Cancer Genome Atlas (TCGA) program provides transcriptomic data for primary cancers across various cancer types. Among multiple datasets, we selected the one including the largest number of patient samples for further analysis. The datasets, except GBM, were downloaded from cBioPortal as there was no normal tissue samples included in the identical dataset with primary tumor of GBM (11). For GBM, the dataset including both primary tumor and normal tissue samples, which was also from TCGA, was downloaded from the GlioVis data portal (12). To avoid any confusion, we revised former phrases into more precise languages.

2. Justification for not using Normal tissue datasets: It will give a clear picture if author analyses the normal tissue dataset which can help to distinguish better between cancer-specific effects.

Response: Thank you for your insightful comment. In the analyses of the expression of the JAG, DLL, and NOTCH families, a comparison between primary tumor and normal tissue samples from the corresponding organs included in the same dataset was performed (Response Figure 15). The results showed that the overall expression of the JAG, DLL, and NOTCH families in tumor samples compared to that in normal tissues was distinct for each cancer type (Response Figure 15A). For BIC and OV, the datasets from TCGA provided only primary tumor samples. 

To provide a clear explanation for the cancer-specific effects, we conducted further analyses to compare the expression of JAG, DLL, and NOTCH families between primary tumors and normal tissues using additional datasets (13, 14). As a result, in BIC, JAG2, DLL3, DLL4, and NOTCH4 were downregulated in tumors compared to normal tissues, whereas other genes showed no significant differences (Response Figure 15B). In contrast, JAG2 and NOTCH3 were upregulated compared to normal tissues, whereas downregulation was observed only in DLL1 in OV (Response Figure 15B).

3. Justification for not evaluating the effects on additional clinical outcomes like treatment response, metastasis, recurrence: Analysis of just single factor like overall survival is a limiting factor of this study and does decrease the generalizability of the study. 

Response: Thank you for your comment. To investigate the relationship between the gene expression of JAG, DLL, and NOTCH family and clinicopathological characteristics, we performed additional analyses comparing primary/metastatic, pre-/post-chemotherapy samples, and primary/recurrent tumors.

First, we analyzed the gene expression of the JAG, DLL, and NOTCH families in primary and metastatic tumors (Response Figure 16). Expression changes in JAG and DLL families varied among genes and cancer types. For example, PRAD metastasis showed increased expression of JAG2, DLL3, and DLL4 whereas that of JAG1 decreased. 

Next, we compared expression between samples from patients before and after therapy (Response Figure 17). Although restricted to BIC and HNSC, the expression of the JAG and DLL families increased, whereas that of the NOTCH family did not change or decrease. In addition, the expression levels of most of the NOTCH family were slightly different whereas those of JAG and DLL families increased in several tumor types, such as CRC, HNSC, OV, PRAD, SCC, and TC.

In addition, we performed an analysis between primary and recurrent tumors from longitudinal biopsies from identical patients with eight tumor species (Response Figure 18-20). There were no differences in the expression of JAG, DLL, and NOTCH families between primary and recurrent GBM, LGG, and HNSC tumors. Meanwhile, the expression of the JAG and DLL families increased in several patients with BIC, LHC, OV, SC, and TC, without a decline. However, the expression of NOTCH family in these patients showed diverse patterns. 

In conclusion, these results suggest that ligand-dependent NOTCH signaling modulation may affect tumor progression, drug resistance, metastasis, and recurrence, which is in line with previous studies (1-3). Additionally, receptor-independent activation of the JAG and DLL families may affect tumor aggressiveness as previously reported (4-6).

4. Explanation on using only mRNA expression data: it would be helpful if author will incorporate proteomic and metabolic data for clear insights.

Response: Thank you for your insightful comment. As you have mentioned, incorporating proteomic data would provide a more accurate understanding of the role of the JAG, DLL, and NOTCH families in cancer owing to their molecular characteristics. The JAG, DLL, and NOTCH family of proteins undergo proteolytic cleavage, forming an intracellular domain that exhibits the transcriptional regulatory functions of target genes (15). These processes depend on ligand-receptor interactions in canonical pathways (1,2). However, in some cases, they are ligand-receptor interaction-independent (4-6). 

Thus, we investigated the protein expression of the JAG, DLL, and NOTCH families comparing primary tumors and normal tissues (Response Figure 17). The analyses included 10 cases (BIC, CRC, GBM, HNSC, KRCC, LGG, LHC, LUAD, OV, and UCEC) among the 15 cancer types, with expression data for both primary tumors and normal tissues. The results demonstrated that the overall expression of the JAG, DLL, and NOTCH families in tumor samples compared to normal tissue was distinguishable in each cancer type. For example, JAG1, NOTCH1, NOTCH2, and NOTCH3 were upregulated in GBM but downregulated in LUAD, except NOTCH3, which is upregulated. In addition, a comparison of primary tumors and normal tissues at the transcriptomic and proteomic levels showed different trends for each cancer type (Response Figure 21 and 22). For example, most JAG, DLL, and NOTCH family genes were upregulated in primary tumors of LUAD compared to normal tissues at both the transcriptomic and proteomic levels. In contrast, the transcriptome and proteome showed opposite expression patterns in GBM. 

As JAG, DLL, and NOTCH family proteins undergo cleavage, translocate into the nucleus, and regulate the transcription of target genes, transcriptional activation might indicate the activation of the JAG, DLL, and NOTCH families (5, 16). Thus, we performed patient survival analysis and demonstrated that DEG enrichment of the JAG, DLL, and NOTCH families was more closely correlated with patient prognosis than gene expression alone (Resp

---

## [Decision Letter · Decision Letter 1]

20 May 2024

PONE-D-23-37633R1Cancer-wide In Silico Analyses Using Differentially Expressed Genes Demonstrate the Functions and Clinical Relevance of JAG, DLL, and NOTCHPLOS ONE

Dear Dr. Kim,

Thank you for submitting your manuscript to PLOS ONE. After careful consideration, we feel that it has merit but does not fully meet PLOS ONE’s publication criteria as it currently stands. Therefore, we invite you to submit a revised version of the manuscript that addresses the points raised during the review process.

We look forward to receiving your revised manuscript.

Kind regards,

Kenji Tanigaki, Ph.D., M.D.

Academic Editor

PLOS ONE

Journal Requirements:

Reviewers' comments:

Reviewer's Responses to Questions

**Comments to the Author**

1. If the authors have adequately addressed your comments raised in a previous round of review and you feel that this manuscript is now acceptable for publication, you may indicate that here to bypass the “Comments to the Author” section, enter your conflict of interest statement in the “Confidential to Editor” section, and submit your "Accept" recommendation.

Reviewer #1: All comments have been addressed

Reviewer #3: All comments have been addressed

2. Is the manuscript technically sound, and do the data support the conclusions?

Reviewer #1: Yes

Reviewer #3: Yes

3. Has the statistical analysis been performed appropriately and rigorously? 

Reviewer #1: N/A

Reviewer #3: Yes

4. Have the authors made all data underlying the findings in their manuscript fully available?

Reviewer #1: No

Reviewer #3: Yes

5. Is the manuscript presented in an intelligible fashion and written in standard English?

Reviewer #1: Yes

Reviewer #3: (No Response)

6. Review Comments to the Author

Reviewer #1: 1.It is suggested that after pooled analysis, the results of JAG, DLL, and NOTCH gene expression and clinicopathological features in common solid tumors can be highlighted and presented separately, not only focusing on GMB, LHC and HNSC.

Reviewer #3: The authors has done a wonderful job while addressing the comments raised in a previous round of review.

In summary, this study is a remarkable contribution to cancer biology, providing extensive insights into the multifaceted roles of Notch signaling. The integration of bioinformatics with experimental data offers a powerful approach to unraveling the complexities of cancer progression and therapeutic resistance. This research not only advances our understanding of Notch signaling but also paves the way for developing more effective cancer treatments.

7. PLOS authors have the option to publish the peer review history of their article (what does this mean?). If published, this will include your full peer review and any attached files.

Reviewer #1: No

Reviewer #3: No

---

## [Author Response · Author response to Decision Letter 1]

29 Jun 2024

Reviewer #1: It is suggested that after pooled analysis, the results of JAG, DLL, and NOTCH gene expression and clinicopathological features in common solid tumors can be highlighted and presented separately, not only focusing on GMB, LHC and HNSC.

Response: Thank you for your insightful comment. The expression of JAG, DLL, and NOTCH families is linked with clinicopathological features including tumor stage progression, metastatic spread, therapeutic response, and recurrence in several tumor types. As DEG enrichment rather than the expression of JAG, DLL, and NOTCH families showed significant correlation with the positive or negative signature of JAG, DLL, and NOTCH families, we performed additional analyses on the linkage between DEG enrichment and clinicopathological phenotypes. The analyses include JAG, DLL, and NOTCH families’ DEG enrichment according to tumor stage, primary/metastasis, pre-/post-chemotherapy, and primary/recurrent status of tumors.

First, we investigated the DEG enrichment of JAG, DLL, and NOTCH families according to the tumor stage. While the mRNA expression of JAG, DLL, and NOTCH families showed a correlation with tumor stage in a few tumor types, DEG enrichment of each families showed a significant correlation in most tumors (Response Figure 1 and 2). These results correspond to DEG enrichment showing a more significant relationship with positive or negative signatures of JAG, DLL, and NOTCH families than mRNA expression of JAG, DLL, and NOTCH families (Response Figure 3).

Interestingly, in GBM, DEG enrichment of ligands except JAG1 and DLL4, which increased stage-dependently, decreased according to tumor stage (Response Figure 2A-E). On the other hand, the enrichment of DEG of all NOTCH receptors showed a stage-dependent increase (Response Figure 3F-I). Especially, NOTCH1-DEG enrichment increased significantly despite the decrease in mRNA expression of NOTCH1 according to tumor stage (Response Figure 1F and 2F). These results might be due to the ligand-dependent signaling activation of NOTCH1 by elevated JAG1 and DLL4 (1, 2). Furthermore, among 1,023 and 1,288 genes up-regulated by transcriptional activity of the intracellular domain of JAG1 and NOTCH1 respectively, 487 genes were commonly regulated by JAG1 and NOTCH1 in our previous study (3). As DEG of Notch ligand and receptor highly overlaps, upregulated JAG1 and DLL4 might have driven these results.

For KRCC, only DLL3-DEG increased stage-dependently while DEG of other ligands and receptors decreased (Response Figure 2). Correspondingly, enrichment of DLL3-DEG is correlated with poor clinical outcome while DEG of other ligands and receptors showed better outcomes (Response Figure 4). In addition, the DEG of JAG and DLL families increased in a stage-dependent manner in LHC (Response Figure 2). These results might indicate the intact relationship between NOTCH signaling and tumor progression in various tumors. Also, various trends of DEG enrichment according to tumor stage for each gene of JAG, DLL, and NOTCH families and tumor type might be due to the diversity of composition and expressions of DEG for each gene and tumor. 

Second, we compared DEG enrichment of JAG, DLL, and NOTCH families in primary and metastatic tumors. While the expression of JAG, DLL, and NOTCH families showed differences in primary and metastatic tumors, DEG enrichment differed only in a few tumors (Response Figures 5, 6 and 7). Although DEG enrichment was more closely related to the tumor stage than the expression, metastasis was not the case. Moreover, although the expression of JAG1 decreased, enrichment of JAG1-DEG increased in metastatic PRAD compared to primary tumors (Response Figures 5A and 6A). As a Notch signal depends on cell-cell interaction, its activation and function are in a context-dependent manner. Interestingly, the expression of transcriptional targets of the Notch signal showed different patterns in primary and metastatic sites in the previous study (4). DEG, which we established, was selected based on separated groups of patients with high or low expression of each JAG, DLL, and NOTCH family gene in primary tumors. Thus, the results on DEG enrichment in metastatic tumors may not be able to represent the context-dependent effect of Notch signal in metastatic tumors.

In addition, we performed an analysis between samples from patients before and after therapy (Response Figures 8 and 9). There were no changes in DEG enrichment in CRC, KRCC, and LHC (Response Figures 9B, 9D, and 9E). For SCC, DEG enrichment of JAG1, JAG2, and NOTCH2 changed after therapy although the expression of all JAG, DLL, and NOTCH families did not show any difference (Response Figures 8F and 9F). The patterns of gene expression and DEG enrichment were similar only in BIC and HNSC. Both the expression and DEG enrichment of JAG1, DLL1, and NOTCH4 significantly increased after therapy in BIC (Response Figures 8C and 9A). JAG2, NOTCH1, and NOTCH3 tend to increase after therapy, although not significant. Interestingly, only DLL3 decreased after therapy in BIC. In HNSC, the expression and DEG enrichment of JAG1, DLL1, and NOTCH2 increased after therapy, while DLL3 and NOTCH4 decreased (Response Figures 8B and 9C). However, the expression of NOTCH3 and its DEG enrichment showed opposite trends. These results might be due to that DEG enrichment can be significantly changed depending on the activation or inhibition of other signals that interact with the Notch signaling after anti-cancer therapy (5).

We also compared DEG enrichment in primary and recurrent tumors (Response Figures 10 and 11). BIC, LGG, and GBM showed a change in DEG enrichment between primary and recurrent tumors. Among Notch ligands, only DLL3-DEG enrichment decreased in recurrent BIC (Response Figure 10). Also, DEG enrichment of JAG1, JAG2, and DLL1 decreased in recurrent GBM (Response Figure 10). In the case of Notch receptors, DEG enrichment of NOTCH1 and NOTCH2 in BIC, NOTCH4 in GBM, and NOTCH2 in LGG decreased in recurrent tumors (Response Figure 11). The outcome of Notch signal activation may differ from crosstalk with multiple signaling pathways in cancer (6). Also, recurrent tumors show dissimilar gene expression and signaling pathway components compared to primary tumors (7). As DEG, which we established, was selected based on primary tumors, DEG enrichment may not be appropriate to explain the context-dependent effect of Notch signal in recurrent tumors.

Collectively, these results indicate that DEG related to Notch signal, in addition to the expression of JAG, DLL, and NOTCH families, might give insights into tumor progression and clinical outcome. In addition, further investigation on the context-dependent change of DEG of JAG, DLL, and NOTCH families might explain the linkage of the Notch signal and pathological phenotypes of cancer.

Reviewer #3: The authors has done a wonderful job while addressing the comments raised in a previous round of review. In summary, this study is a remarkable contribution to cancer biology, providing extensive insights into the multifaceted roles of Notch signaling. The integration of bioinformatics with experimental data offers a powerful approach to unraveling the complexities of cancer progression and therapeutic resistance. This research not only advances our understanding of Notch signaling but also paves the way for developing more effective cancer treatments.

Response: Thank you very much for your kind review. In the present study, we demonstrated that JAG, DLL, and NOTCH families gene expression and their DEG vary among tumor types and that they might regulate diverse cellular physiology. We also investigated the cancer-wide function of JAG, DLL, and NOTCH families and discussed the linkage with clinicopathological features including tumor progression, metastasis, therapeutic response, and therapy resistance in tumors. As you mentioned, our results may provide insights into the diverse role of Notch signaling in tumor progression and therapeutic resistance. A comprehensive investigation of Notch signaling and associated biological features would support further understanding of the regulatory mechanism of malignancy and, therefore might provide the way to the most efficient therapeutic approach.

---

## [Editor Report · Decision Letter 2]

16 Jul 2024

Cancer-wide In Silico Analyses Using Differentially Expressed Genes Demonstrate the Functions and Clinical Relevance of JAG, DLL, and NOTCH

PONE-D-23-37633R2

Dear Dr. Kim,

We’re pleased to inform you that your manuscript has been judged scientifically suitable for publication and will be formally accepted for publication once it meets all outstanding technical requirements.

Kind regards,

Kenji Tanigaki, Ph.D., M.D.

Academic Editor

PLOS ONE

---

## [Editor Report · Acceptance letter]

19 Jul 2024

PONE-D-23-37633R2 

PLOS ONE

Dear Dr. Kim, 

I'm pleased to inform you that your manuscript has been deemed suitable for publication in PLOS ONE. Congratulations! Your manuscript is now being handed over to our production team.

Kind regards, 

on behalf of

Dr. Kenji Tanigaki 

Academic Editor

PLOS ONE